# Soft Contrastive Learning for Time Series

**Seunghan Lee, Taeyoung Park, Kibok Lee**
Department of Statistics and Data Science, Yonsei University
{seunghan9613,tpark,kibok}@yonsei.ac.kr

## Abstract

Contrastive learning has shown to be effective to learn representations from time series in a self-supervised way. However, contrasting similar time series instances or values from adjacent timestamps within a time series leads to ignore their inherent correlations, which results in deteriorating the quality of learned representations. To address this issue, we propose *SoftCLT*, a simple yet effective soft contrastive learning strategy for time series. This is achieved by introducing instance-wise and temporal contrastive loss with soft assignments ranging from zero to one. Specifically, we define soft assignments for 1) instance-wise contrastive loss by the distance between time series on the data space, and 2) temporal contrastive loss by the difference of timestamps. SoftCLT is a plug-and-play method for time series contrastive learning that improves the quality of learned representations without bells and whistles. In experiments, we demonstrate that SoftCLT consistently improves the performance in various downstream tasks including classification, semi-supervised learning, transfer learning, and anomaly detection, showing state-of-the-art performance. Code is available at this repository: https://github.com/seunghan96/softclt.

## 1 Introduction

Time series (TS) data are ubiquitous in many fields, including finance, energy, healthcare, and transportation (Ding et al., 2020; Lago et al., 2018; Solares et al., 2020; Cai et al., 2020). However, annotating TS data can be challenging as it often requires significant domain expertise and time. To overcome the limitation and utilize unlabeled data without annotations, self-supervised learning has emerged as a promising representation learning approach not only in natural language processing (Devlin et al., 2018; Gao et al., 2021) and computer vision (Chen et al., 2020; Dosovitskiy et al., 2021), but also in TS analysis (Franceschi et al., 2019; Yue et al., 2022). In particular, contrastive learning (CL) has demonstrated remarkable performance across different domains (Chen et al., 2020; Gao et al., 2021; Yue et al., 2022). As it is challenging to determine similarities of instances in self-supervised learning, recent CL works apply data augmentation to generate two views per data and take views from the same instance as positive pairs and the others as negatives (Chen et al., 2020). However, we argue that the standard CL objective might be harmful for TS representation learning, because inherent correlations in similar TS instances and values nearby timestamps within a TS, which could be a strong self-supervision, are ignored in CL. For example, distance metrics such as dynamic time warping (DTW) have been widely used for measuring the similarities of TS data, and contrasting TS data might lose such information. Also, values with close timestamps are usually similar in natural TS data, so contrasting all values with different timestamps with the same degree of penalty as in previous CL methods (Eldele et al., 2021; Yue et al., 2022) might not be optimal. Motivated by this, we explore the following research question: *how can we take account of the similarities of time series data for better contrastive representation learning?* To this end, we propose **Soft** **C**ontrastive **L**earning for **T**ime series (*SoftCLT*). Specifically, we propose to consider the InfoNCE loss (Oord et al., 2018) not only for the positive pairs but also all other pairs and compute their weighted summation in both instance-wise CL and temporal CL, where instance-wise CL contrasts the representations of TS instances, while temporal CL contrasts the representations of timestamps within a single TS, as shown in Figure 1. We propose to assign soft assignments based on the distance between TS for the instance-wise CL, and the difference of timestamps for the temporal CL. This formulation can be seen as a generalization of the standard contrastive loss, as the proposed loss becomes the contrastive loss if we replace soft assignments with hard assignments of either zero for negative or one for positive.

| | T-Loss (NeurIPS 2019) | Self-Time (arxiv 2020) | TNC (ICLR 2021) | TS-SD (IJCNN 2021) | TS-TCC (IJCAI 2021) | TS2Vec (AAAI 2022) | Mixing-Up (PR Letters 2022) | CoST (ICLR 2022) | TimeCLR (KBS 2022) | TF-C (NeurIPS 2022) | CA-TCC (TPAMI 2023) | SoftCLT (Ours) |
|---|---|---|---|---|---|---|---|---|---|---|---|---|
| Instanse-wise CL | ✓ | ✓ | | ✓ | | ✓ | ✓ | ✓ | ✓ | ✓ | | ✓ |
| Temporal CL | | ✓ | ✓ | | ✓ | ✓ | | | | | ✓ | ✓ |
| Hierarchical CL | | | | | | ✓ | | | | | | ✓ |
| Soft CL | | | | | | | | | | | | ✓ |

Table 1: Comparison table of contrastive learning methods in time series.

We conduct extensive experiments in various tasks, including TS classification, semi-supervised classification, transfer learning, and anomaly detection tasks to prove the effectiveness of the proposed method. Experimental results validate that our method improves the performance of previous CL methods, achieving state-of-the-art (SOTA) performance on a range of downstream tasks. The main contributions of this paper are summarized as follows:

- We propose SoftCLT, a simple yet effective soft contrastive learning strategy for TS. Specifically, we propose soft contrastive losses for instance and temporal dimensions, respectively, to address limitations of previous CL methods for TS.

- We provide extensive experimental results on various tasks for TS, showing that our method improves SOTA performance on a range of downstream tasks. For example, SoftCLT improves the average accuracy of 125 UCR datasets and 29 UEA datasets by 2.0% and 3.9%, respectively, compared to the SOTA unsupervised representation for classification tasks.

- SoftCLT is easily applicable to other CL frameworks for TS by introducing soft assignments and its overhead is negligible, making it practical for use.

## 2    RELATED WORK

**Self-supervised learning.**  In recent years, self-supervised learning has gained lots of attention for its ability to learn powerful representations from large amounts of unlabeled data. Self-supervised learning is done by training a model to solve a pretext task derived from a certain aspect of data without supervision. As a self-supervised pretext task, next token prediction (Brown et al., 2020) and masked token prediction (Devlin et al., 2018) are commonly used in natural language processing, while solving jigsaw puzzles (Noroozi & Favaro, 2016) and rotation prediction (Gidaris & Komodakis, 2018) are proposed in computer vision. In particular, contrastive learning (Hadsell et al., 2006) has shown to be an effective pretext task across domains, which maximizes similarities of positive pairs while minimizing similarities of negative pairs (Gao et al., 2021; Chen et al., 2020; Yue et al., 2022).

**Contrastive learning in time series.**  In the field of TS analysis, several designs for positive and negative pairs have been proposed for CL, taking into account the invariant properties of TS. Table 1 compares various CL methods in TS including ours in terms of several properties. T-Loss (Franceschi et al., 2019) samples a random subseries from a TS and treats them as positive when they belong to its subseries, and negative if belong to subseries of other TS. Self-Time (Fan et al., 2020) captures inter-sample relation between TS by defining augmented sample of same TS as positive and negative otherwise, and captures intra-temporal relation within TS by solving a classification task, where the class labels are defined using the temporal distance between the subseries. TNC (Tonekaboni et al., 2021) defines temporal neighborhood of windows using normal distribution and treats samples in neighborhood as positives. TS-SD (Shi et al., 2021) trains a model using triplet similarity discrimination task, where the goal is to identify which of two TS is more similar to a given TS, using DTW to define similarity. TS-TCC (Eldele et al., 2021) proposes a temporal contrastive loss by making the augmentations predict each other's future, and CA-TCC (Eldele et al., 2023), which is the extension of TS-TCC to the semi-supervised setting, adopts the same loss. TS2Vec (Yue et al., 2022) splits TS into two subseries and defines hierarchical contrastive loss in both instance and temporal dimensions. Mixing-up (Wickstrøm et al., 2022) generates new TS by mixing two TS, where the goal is to predict the mixing weights. CoST (Woo et al., 2022) utilizes both time domain and frequency domain contrastive losses to learn disentangled seasonal-trend representations of TS. TimeCLR (Yang et al., 2022) introduces phase-shift and amplitude change augmentations, which are data augmentation methods based on DTW. TF-C (Zhang et al., 2022) learns both time- and frequency-based representations of TS and proposes a novel time-frequency consistency architecture. In the medical domain, Subject-Aware CL (Cheng et al., 2020) proposes an instance-wise CL framework where the temporal information is entangled by architecture design, and CLOCS (Kiyasseh et al., 2021) proposes to consider spatial dimension specifically available in their application, which is close to the channels in general TS. While previous CL methods for TS compute *hard* contrastive loss, where the similarities between all negative pairs are equally minimized, we introduce *soft* contrastive loss for TS.

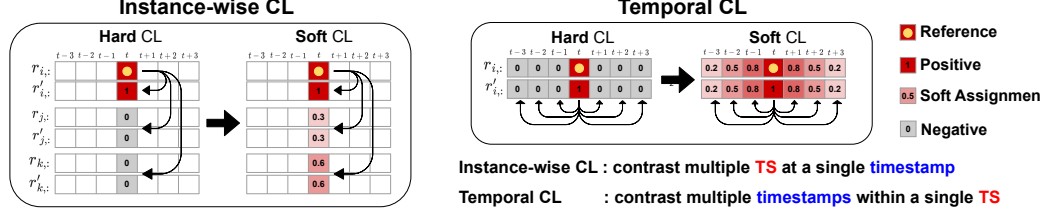

Figure 1: **Overall framework of SoftCLT.** Unlike the conventional hard CL that gives either positive or negative assignments to sample pairs, SoftCLT gives soft assignments to both instance-wise and temporal relationships. Two views of the same sample are denoted as $r$ and $\tilde{r}$, respectively.

**Soft contrastive learning.** CL is typically done by batch instance discrimination, where each instance is considered to be in a distinct class. However, this approach can pose a risk of pushing similar samples farther apart in the embedding space. To address this issue, several methods have been proposed, including a method that utilizes soft assignments of images (Thoma et al., 2020) based on feature distances and geometric proximity measures. NNCLR (Dwibedi et al., 2021) defines additional positives for each view by extracting top-$k$ neighbors in the feature space. NCL (Yèche et al., 2021) finds neighbors using supervision from the medical domain knowledge and jointly optimizes two conflicting losses with a trade-off: the neighbor alignment loss maximizing the similarity of neighbors as well as positive pairs, and the neighbor discriminative loss maximizing the similarity of positive pairs while minimizing the similarity of neighbors. SNCLR (Ge et al., 2023), which extends NNCLR with soft assignments, employs an attention module to determine the correlations between the current and neighboring samples. CO2 (Wei et al., 2021) introduces consistency regularization to enforce relative distribution consistency between different positive views and all negatives, resulting in soft relationships between samples. ASCL (Feng & Patras, 2022) introduces soft inter-sample relations by transforming the original instance discrimination task into a multi-instance soft discrimination task. Previous soft CL methods in non-TS domains compute soft assignments on the *embedding space*, because similarities of instances on the data space are difficult to measure, particularly in computer vision (Chen et al., 2020). In contrast, we propose to compute soft assignments based on the distance between TS instances on the *data space*.

**Masked modeling in time series.** Other than CL, masked modeling has recently been studied as a pretext task for self-supervised learning in TS by masking out a portion of TS and predicting the missing values. While CL has demonstrated remarkable performance in high-level classification tasks, masked modeling has excelled in low-level forecasting tasks (Dong et al., 2023; Huang et al., 2022; Xie et al., 2022). TST (Zerveas et al., 2021) adopts the masked modeling paradigm to TS, where the goal is to reconstruct the masked timestamps. PatchTST (Nie et al., 2023) aims to predict the masked subseries-level patches to capture the local semantic information and reduce memory usage. SimMTM (Dong et al., 2023) reconstructs the original TS from multiple masked TS.

## 3 METHODOLOGY

In this section, we propose SoftCLT by introducing soft assignments to instance-wise and temporal contrastive losses to capture both inter-sample and intra-temporal relationships, respectively. For instance-wise CL, we use distance between TS on the data space to capture the inter-sample relations, and for temporal CL, we use the difference between timestamps to consider the temporal relation within a single TS. The overall framework of SoftCLT is illustrated in Figure 1.

### 3.1 PROBLEM DEFINITION

This paper addresses the task of learning a nonlinear embedding function $f_\theta : x \to r$, given a batch of $N$ time series $\mathcal{X} = \{x_1, \dots, x_N\}$. Our goal is to learn $f_\theta$ mapping a time series $x_i \in \mathbb{R}^{T \times D}$ to a representation vector $r_i = [r_{i,1}, \dots, r_{i,T}]^\top \in \mathbb{R}^{T \times M}$, where $T$ is the sequence length, $D$ is the input feature dimension, and $M$ is the embedded feature dimension.

### 3.2 SOFT INSTANCE-WISE CONTRASTIVE LEARNING

Contrasting all instances within a batch might be harmful for TS representation learning because similar instances are learned to be far away from each other on the embedding space. Unlike other domains such as computer vision, the distance between TS data computed on the data space

is useful for measuring the similarity of them. For example, the pixel-by-pixel distance of two different images is not related to their similarities in general, that of two TS data is useful to measure their similarities. With a min-max normalized distance metric $D(\cdot, \cdot)$, we define a soft assignment for a pair of data indices $(i, i')$ for the instance-wise contrastive loss using the sigmoid function $\sigma(a) = 1/(1 + \exp(-a))$:

$$w_I(i, i') = 2\alpha \cdot \sigma\left(-\tau_I \cdot D(x_i, x_{i'})\right), \tag{1}$$

where $\tau_I$ is a hyperparameter controlling the sharpness and $\alpha$ is the upper bound in the range of $[0, 1]$ to distinguish pairs of the same TS and pairs of different TS close to each other; when $\alpha = 1$, we give the assignment of one to the pairs with the distance of zero as well as the pairs of the same TS. Note that distances between TS are computed with the *original* TS rather than the augmented views, because the pairwise distance matrix can be precomputed offline or cached for efficiency.

For the choice of the distance metric $D$, we conduct an ablation study in Table 6d, comparing 1) cosine distance, 2) Euclidean distance, 3) dynamic time warping (DTW), and 4) time alignment measurement (TAM) (Folgado et al., 2018). Among them, we choose DTW as the distance metric throughout the experiments based on the result in Table 6d. While the computational complexity of DTW is $\mathcal{O}(T^2)$ for two TS of length $T$ which might be costly for large-scale datasets, it can be precomputed offline or cached to facilitate efficient calculations, or its fast version such as FastDTW (Salvador & Chan, 2007) with the complexity of $\mathcal{O}(T)$ can be used. We empirically confirmed that the output of DTW and FastDTW is almost the same, such that the CL results also match.

Let $r_{i,t} = r_{i+2N,t}$ and $\tilde{r}_{i,t} = r_{i+N,t}$ be the embedding vectors from two augmentations of $x_i$ at timestamp $t$ for conciseness. Inspired by the fact that the contrastive loss can be interpreted as the cross-entropy loss (Lee et al., 2021), we define a softmax probability of the relative similarity out of all similarities considered when computing the loss as:

$$p_I((i, i'), t) = \frac{\exp(r_{i,t} \circ r_{i',t})}{\sum_{j=1, j \neq i}^{2N} \exp(r_{i,t} \circ r_{j,t})}, \tag{2}$$

where we use the dot product as the similarity measure $\circ$. Then, the soft instance-wise contrastive loss for $x_i$ at timestamp $t$ is defined as:

$$\ell_I^{(i,t)} = -\log p_I((i, i+N), t) - \sum_{j=1, j \neq \{i, i+N\}}^{2N} w_I(i, j \bmod N) \cdot \log p_I((i, j), t). \tag{3}$$

The first term in $\ell_I^{(i,t)}$ corresponds to the loss of the positive pair, and the second term corresponds to that of the other pairs weighted by soft assignments $w_I(i, i')$. Note that this loss can be seen as a generalization of the hard instance-wise contrastive loss, which is the case when $\forall w_I(i, i') = 0$.

### 3.3 SOFT TEMPORAL CONTRASTIVE LEARNING

Following the intuition that values in adjacent timestamps are similar, we propose to compute a soft assignment based on the difference between timestamps for temporal contrastive loss. Similar to the soft instance-wise contrastive loss, the assignment is close to one when timestamps get closer and zero when they get farther away. We define a soft assignment for a pair of timestamps $(t, t')$ for the temporal contrastive loss as:

$$w_T(t, t') = 2 \cdot \sigma\left(-\tau_T \cdot |t - t'|\right), \tag{4}$$

where $\tau_T$ is a hyperparameter controlling the sharpness. As the degree of closeness between timestamps varies across datasets, we tune $\tau_T$ to control the degree of soft assignments. Figure 2a illustrates an example of soft assignments with respect to timestamp difference with different $\tau_T$.

**Hierarchical loss.** For temporal CL, we consider hierarchical contrasting on intermediate representations in the network $f_\theta$ as done in prior CL methods for TS. Specifically, we adopt the hierarchical contrastive loss proposed in TS2Vec (Yue et al., 2022), where the losses are computed on intermediate representations after each max-pooling layer along the temporal axis and then aggregated. As shown in Figure 2b, similarities between adjacent time step decrease after pooling, we adjust $\tau_T$ by multiplying $m^k$ in Eq. 4, i.e., $\tau_T = m^k \cdot \tilde{\tau}_T$ where $m$ is the kernel size of pooling layers, $k$ is the depth, and $\tilde{\tau}_T$ is the base hyperparameter.

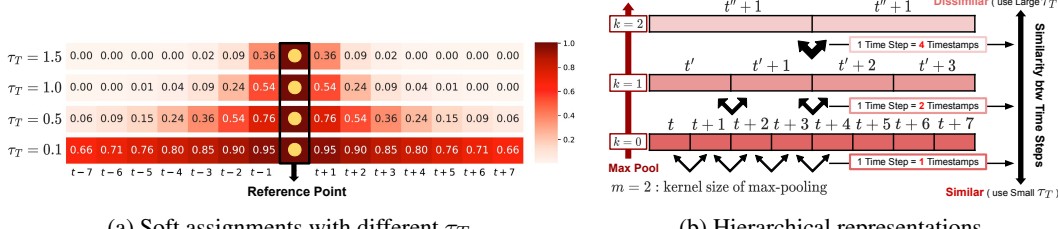

(a) Soft assignments with different $\tau_T$.  (b) Hierarchical representations.

Figure 2: (a) shows examples of soft assignments for soft temporal CL, where a smaller $\tau_T$ results in smoother assignments. (b) is an example of hierarchical representations, demonstrating that increasing layer depth results in a larger semantic difference between adjacent time steps, so $\tau_T$ should be increased to compensate for it.

Now, let $r_{i,t} = r_{i,t+2T}$ and $\tilde{r}_{i,t} = r_{i,t+T}$ be the embedding vectors from two augmentations of $x_i$ at timestamp $t$ for conciseness. Similar to Eq. 2, we define a softmax probability of the relative similarity out of all similarities considered when computing the loss as:

$$p_T(i, (t, t')) = \frac{\exp(r_{i,t} \circ r_{i,t'})}{\sum_{s=1, s \neq t}^{2T} \exp(r_{i,t} \circ r_{i,s})}. \tag{5}$$

Then, the soft temporal contrastive loss for $x_i$ at timestamp $t$ is defined as:

$$\ell_T^{(i,t)} = -\log p_T(i, (t, t+T)) - \sum_{s=1, s \neq \{t, t+T\}}^{2T} w_T(t, s \bmod T) \cdot \log p_T(i, (t, s)). \tag{6}$$

Similar to the soft instance-wise contrastive loss, this loss can be seen as a generalization of the hard temporal contrastive loss, which is the case when $\forall w_T(t, t') = 0$.

The final loss for SoftCLT is the joint of the soft instance-wise and temporal contrastive losses:

$$\mathcal{L} = \frac{1}{4NT} \sum_{i=1}^{2N} \sum_{t=1}^{2T} (\lambda \cdot \ell_I^{(i,t)} + (1 - \lambda) \cdot \ell_T^{(i,t)}), \tag{7}$$

where $\lambda$ is a hyperparameter controlling the contribution of each loss, set to 0.5 unless specified. The proposed loss has an interesting mathematical interpretation that it can be seen as the scaled KL divergence of the softmax probabilities from the normalized soft assignments, where the scale is the sum of soft assignments. We provide more details in Appendix D.

## 4 EXPERIMENTS

We conduct extensive experiments to validate the proposed method and assess its performance in different tasks: (1) **classification** with univariate and multivariate TS, (2) **semi-supervised classification** by (i) self-supervised learning followed by fine-tuning and (ii) semi-supervised learning, (3) **transfer learning** in in-domain and cross-domain scenarios, and (4) **anomaly detection** in normal and cold-start settings. We also conduct ablation studies to validate the effectiveness of SoftCLT as well as its design choices. Finally, we visualize pairwise distance matrices and t-SNE (Van der Maaten & Hinton, 2008) of temporal representations to show the effect of SoftCLT over previous methods. We use the data augmentation strategies of the methods we apply our SoftCLT to: TS2Vec generates two views as TS segments with overlap, and TS-TCC/CA-TCC generate two views with weak and strong augmentations, using the jitter-and-scale and permutation-and-jitter strategies, respectively.

### 4.1 CLASSIFICATION

We conduct experiments on TS classification tasks with $125^1$ UCR archive datasets (Dau et al., 2019) for univariate TS and $29^2$ UEA archive datasets (Bagnall et al., 2018) for multivariate TS,

---

[1] Some of the previous methods cannot handle missing observations, so three of the 128 datasets are omitted.
[2] One of the 30 datasets is omitted for a fair comparison with some of the previous methods.

| Method | 125 UCR datasets | | 29 UEA datasets | |
|---|---|---|---|---|
| | Avg. Acc.(%) | Avg. Rank | Avg. Acc.(%) | Avg. Rank |
| DTW-D | 72.7 | 5.30 | 65.0 | 4.60 |
| TNC | 76.1 | 4.42 | 67.7 | 4.76 |
| TST | 64.1 | 6.19 | 63.5 | 5.26 |
| TS-TCC | 75.7 | 4.29 | 68.2 | 4.38 |
| T-Loss | 80.6 | 3.50 | 67.5 | 3.86 |
| TS2Vec | 83.0 | 2.80 | 71.2 | 3.28 |
| + Ours | 85.0(+ 2.0) | 1.49 | 75.1(+ 3.9) | 1.86 |

Table 2: Accuracy and rank on UCR/UEA.

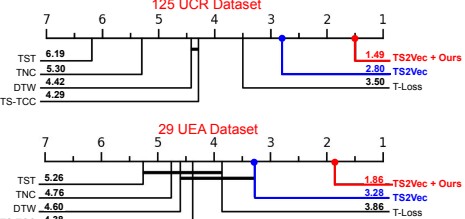

Figure 3: CD diagram on UCR/UEA.

| | 1% of labeled data | | | | | | | | | | | | |
|---|---|---|---|---|---|---|---|---|---|---|---|---|---|
| | Self-supervised learning | | | | | | | Semi-supervised learning | | | | | |
| Dataset | SSL-ECG | CPC | SimCLR | TS2Vec | + Ours | TS-TCC | + Ours | Mean-Teacher | DivideMix | SemiTime | FixMatch | CA-TCC | + Ours |
| HAR | 60.0 / 54.0 | 65.4 / 63.8 | 65.8 / 64.3 | 88.6 / 88.5 | 91.0 / 91.0 | 70.5 / 69.5 | 82.9 / 82.8 | 75.9 / 74.0 | 76.5 / 75.4 | 77.6 / 76.3 | 76.4 / 75.6 | 77.3 / 76.2 | 90.6 / 90.6 |
| Epilepsy | 89.3 / 86.0 | 88.9 / 85.8 | 88.3 / 84.0 | 95.8 / 93.4 | 96.3 / 94.1 | 91.2 / 89.2 | 95.6 / 95.6 | 91.5 / 90.6 | 90.9 / 89.4 | 91.6 / 90.8 | 93.2 / 92.2 | 92.0 / 91.9 | 97.9 / 97.9 |
| Wafer | 93.4 / 76.1 | 93.5 / 78.4 | 93.8 / 78.5 | 67.9 / 56.1 | 95.3 / 88.1 | 93.2 / 76.7 | 96.5 / 96.5 | 94.7 / 84.7 | 93.2 / 82.0 | 94.4 / 84.4 | 95.0 / 84.8 | 95.1 / 85.1 | 98.9 / 98.8 |
| FordA | 67.9 / 66.2 | 75.8 / 75.2 | 55.9 / 55.7 | 86.4 / 86.4 | 87.1 / 87.1 | 80.6 / 80.0 | 81.5 / 81.2 | 71.7 / 71.5 | 73.7 / 73.3 | 75.1 / 74.4 | 74.5 / 74.3 | 82.3 / 81.7 | 90.6 / 90.5 |
| FordB | 64.4 / 60.5 | 66.8 / 65.0 | 50.9 / 49.8 | 65.4 / 65.4 | 67.9 / 67.9 | 78.6 / 78.6 | 74.8 / 74.8 | 65.9 / 65.8 | 54.5 / 54.1 | 67.6 / 67.5 | 56.7 / 55.4 | 73.8 / 73.0 | 78.3 / 78.2 |
| POC | 62.5 / 41.2 | 64.8 / 48.2 | 61.5 / 38.4 | 63.1 / 62.8 | 63.6 / 62.8 | 63.8 / 48.1 | 65.4 / 64.6 | 62.1 / 40.8 | 62.1 / 40.7 | 62.0 / 40.4 | 61.9 / 40.0 | 63.4 / 49.3 | 73.3 / 71.7 |
| StarLightCurves | 78.3 / 72.0 | 80.8 / 74.4 | 80.6 / 71.6 | 82.9 / 60.6 | 85.6 / 62.9 | 86.0 / 79.2 | 86.0 / 79.3 | 79.4 / 77.7 | 79.0 / 77.2 | 79.5 / 77.8 | 77.2 / 71.6 | 85.8 / 77.8 | 94.1 / 94.2 |
| ElectricDevices | 60.1 / 50.0 | 59.3 / 48.9 | 62.5 / 51.2 | 57.6 / 48.6 | 62.0 / 53.0 | 63.6 / 56.4 | 64.6 / 63.2 | 48.9 / 48.3 | 59.8 / 49.4 | 57.3 / 48.1 | 58.2 / 46.9 | 65.9 / 56.7 | 70.3 / 68.8 |
| | 5% of labeled data | | | | | | | | | | | | |
| | Self-supervised learning | | | | | | | Semi-supervised learning | | | | | |
| HAR | 63.7 / 58.6 | 75.4 / 74.7 | 75.8 / 74.9 | 91.1 / 91.0 | 92.1 / 92.1 | 77.6 / 76.7 | 92.6 / 92.6 | 88.2 / 88.1 | 89.1 / 89.1 | 87.6 / 87.1 | 87.6 / 87.3 | 88.3 / 88.3 | 91.4 / 91.4 |
| Epilepsy | 92.8 / 89.0 | 92.8 / 90.2 | 91.3 / 89.2 | 96.0 / 93.6 | 96.7 / 94.9 | 93.1 / 93.7 | 96.2 / 96.1 | 94.0 / 93.6 | 93.9 / 93.4 | 94.0 / 93.0 | 93.7 / 92.4 | 94.5 / 94.0 | 98.0 / 97.9 |
| Wafer | 94.9 / 84.5 | 92.5 / 79.4 | 94.8 / 83.3 | 98.8 / 96.9 | 98.8 / 96.8 | 93.2 / 81.2 | 98.2 / 98.2 | 94.4 / 83.8 | 94.7 / 84.6 | 95.0 / 84.7 | 94.9 / 84.4 | 95.8 / 85.2 | 98.9 / 98.8 |
| FordA | 73.6 / 70.7 | 86.5 / 86.5 | 69.6 / 68.9 | 91.2 / 91.2 | 92.5 / 92.5 | 89.9 / 89.9 | 93.2 / 93.2 | 82.6 / 82.5 | 84.0 / 83.9 | 83.8 / 83.7 | 83.8 / 83.8 | 90.9 / 90.8 | 93.3 / 93.3 |
| FordB | 71.7 / 69.8 | 86.3 / 86.2 | 63.0 / 60.7 | 74.9 / 74.9 | 78.8 / 78.6 | 86.1 / 85.9 | 88.0 / 88.0 | 64.6 / 62.7 | 60.2 / 57.9 | 65.0 / 62.6 | 62.7 / 60.7 | 88.2 / 88.2 | 89.4 / 89.4 |
| POC | 62.9 / 43.3 | 66.9 / 44.3 | 62.7 / 42.4 | 70.4 / 68.0 | 70.9 / 69.7 | 62.6 / 42.6 | 69.4 / 66.3 | 62.1 / 41.2 | 62.9 / 45.9 | 62.4 / 41.8 | 63.1 / 43.6 | 66.2 / 52.8 | 73.1 / 70.7 |
| StarLightCurves | 82.6 / 74.5 | 89.1 / 84.5 | 84.2 / 74.0 | 90.0 / 87.6 | 92.3 / 89.8 | 89.6 / 82.7 | 86.2 / 85.5 | 84.9 / 83.9 | 85.6 / 84.1 | 84.6 / 83.8 | 84.1 / 77.5 | 88.8 / 81.1 | 94.3 / 94.2 |
| ElectricDevices | 63.7 / 56.1 | 62.4 / 58.1 | 63.9 / 58.6 | 62.9 / 54.7 | 62.4 / 54.4 | 65.1 / 59.2 | 65.1 / 63.8 | 70.1 / 60.9 | 72.0 / 62.1 | 71.6 / 61.1 | 62.6 / 55.5 | 66.4 / 59.3 | 70.6 / 68.9 |

Table 3: **Semi-supervised classification results.** The table shows the results of fine-tuning self- and semi-supervised models, with 1% and 5% of labels. **Best results** across each dataset are in bold, while the second-best results are underlined. The accuracy and MF1 score are reported in order.

respectively. Specifically, we apply SoftCLT to TS2Vec (Yue et al., 2022), which has demonstrated SOTA performance on the above datasets. As baseline methods, we consider DTW-D (Chen et al., 2013), TNC (Tonekaboni et al., 2021), TST (Zerveas et al., 2021), TS-TCC (Eldele et al., 2021), T-Loss (Franceschi et al., 2019), and TS2Vec (Yue et al., 2022). The experimental protocol follows that of T-Loss and TS2Vec, where the SVM classifier with the RBF kernel is trained on top of the instance-level representations obtained by max-pooling representations of all timestamps. Table 2 and the critical difference (CD) diagram based on the Wilcoxon-Holm method (Ismail Fawaz et al., 2019) shown in Figure 3 demonstrate that the proposed method improves SOTA performance by a significant margin on both datasets in terms of accuracy and rank. In Figure 3, the best and second-best results for each dataset are in red and blue, respectively. We also connect methods with a bold line if their difference is not statistically significant in terms of the average rank with a confidence level of 95%, which shows that the performance gain by the proposed method is significant.

## 4.2 SEMI-SUPERVISED CLASSIFICATION

We conduct experiments on semi-supervised classification tasks by adopting SoftCLT to TS-TCC (Eldele et al., 2021) and its extension CA-TCC (Eldele et al., 2023), which are the methods that incorporate CL into self- and semi-supervised learning, respectively. As baseline methods, we consider SSL-ECG (Sarkar & Etemad, 2020), CPC (Oord et al., 2018), SimCLR (Chen et al., 2020) and TS-TCC (Eldele et al., 2021) for self-supervised learning, and Mean-Teacher (Tarvainen & Valpola, 2017), DivideMix (Li et al., 2020), SemiTime (Fan et al., 2021), FixMatch (Sohn et al., 2020) and CA-TCC (Eldele et al., 2023) for semi-supervised learning. Note that both TS-TCC and CA-TCC perform instance-wise and temporal contrasting, however, their temporal contrasting is achieved by predicting one view's future from another, which is different from the conventional contrastive loss with positive and negative pairs. Therefore, we adopt our soft temporal contrastive loss as an additional loss to both methods. For evaluation, we utilize the same experimental settings and datasets of CA-TCC, which includes eight datasets (Anguita et al., 2013; Andrzejak et al., 2001; Dau et al., 2019), six of which are from the UCR archive. We consider two semi-supervised learning scenarios, (1) self-supervised learning with unlabeled data followed by supervised fine-tuning with labeled data and (2) semi-supervised learning with both labeled and unlabeled data, following CA-TCC (Eldele et al., 2023). Table 3 presents the experimental results with both methods in scenarios with 1% and 5% labeled datasets, showing that applying SoftCLT achieves the best overall performance across most of the datasets in both scenarios.

| | In-domain transfer learning | | | | Cross-domain transfer learning | | | | | | | | | | | |
| | SleepEEG → Epilepsy | | | | SleepEEG → FD-B | | | | SleepEEG → Gesture | | | | SleepEEG → EMG | | | |
| | ACC. | PRE. | REC. | F₁ | ACC. | PRE. | REC. | F₁ | ACC. | PRE. | REC. | F₁ | ACC. | PRE. | REC. | F₁ |
|---|---|---|---|---|---|---|---|---|---|---|---|---|---|---|---|---|
| TS-SD | 89.52 | 80.18 | 76.47 | 77.67 | 55.66 | 57.10 | 60.54 | 57.03 | 69.22 | 66.98 | 68.67 | 66.56 | 46.06 | 15.45 | 33.33 | 21.11 |
| TS2Vec | 93.95 | 90.59 | 90.39 | 90.45 | 47.90 | 43.39 | 48.42 | 43.89 | 69.17 | 65.45 | 68.54 | 65.70 | 78.54 | 80.40 | 67.85 | 67.66 |
| Mixing-Up | 80.21 | 40.11 | 50.00 | 44.51 | 67.89 | 71.46 | 76.13 | 72.73 | 69.33 | 67.19 | 69.33 | 64.97 | 30.24 | 10.99 | 25.83 | 15.41 |
| CLOCS | 95.07 | 93.01 | 91.27 | 92.06 | 49.27 | 48.24 | 58.73 | 47.46 | 44.33 | 42.37 | 44.33 | 40.14 | 69.85 | 53.06 | 53.54 | 51.39 |
| CoST | 88.40 | 88.20 | 72.34 | 76.88 | 47.06 | 38.79 | 38.42 | 34.79 | 68.33 | 65.30 | 68.33 | 66.42 | 53.65 | 49.07 | 42.10 | 35.27 |
| LaST | 86.46 | 90.77 | 66.35 | 70.67 | 46.67 | 43.90 | 47.71 | 45.17 | 64.17 | 70.36 | 64.17 | 58.76 | 66.34 | 79.34 | 63.33 | 72.55 |
| TF-C | 94.95 | 94.56 | 89.08 | 91.49 | 69.38 | 75.59 | 72.02 | 74.87 | 76.42 | 77.31 | 74.29 | 75.72 | 81.71 | 72.65 | 81.59 | 76.83 |
| TST | 80.21 | 40.11 | 50.00 | 44.51 | 46.40 | 41.58 | 45.50 | 41.34 | 69.17 | 66.60 | 69.17 | 66.01 | 46.34 | 15.45 | 33.33 | 21.11 |
| SimMTM | 95.49 | 93.36 | 92.28 | 92.81 | 69.40 | 74.18 | 76.41 | 75.11 | 80.00 | 79.03 | 80.00 | 78.67 | 97.56 | 98.33 | 98.04 | 98.14 |
| TS-TCC | 92.53 | 94.51 | 81.81 | 86.33 | 54.99 | 52.79 | 63.96 | 54.18 | 71.88 | 71.35 | 71.67 | 69.84 | 78.89 | 58.51 | 63.10 | 59.04 |
| + Ours | **97.00** | **97.07** | **97.00** | **96.92** | **80.45** | **86.84** | **85.68** | **85.48** | **95.00** | **95.59** | **95.00** | **95.12** | **100** | **100** | **100** | **100** |

(a) Transfer learning under in- and cross-domain scenarios.

| | A → B | A → C | A → D | B → A | B → C | B → D | C → A | C → B | C → D | D → A | D → B | D → C | Avg |
|---|---|---|---|---|---|---|---|---|---|---|---|---|---|
| Supervised | 34.38 | 44.94 | 34.57 | 52.93 | 63.67 | 99.82 | 52.93 | 84.02 | 83.54 | 53.15 | 99.56 | 62.43 | 63.8 |
| TS-TCC | 43.15 | 51.50 | 42.74 | 47.98 | 70.38 | 99.30 | 38.89 | **98.31** | **99.38** | 51.91 | **99.96** | 70.31 | 67.82 |
| + Ours | **76.83** | **74.35** | **78.34** | **53.37** | **75.11** | **99.38** | **53.26** | 85.59 | 86.29 | **53.30** | 93.55 | **70.93** | **75.03** (+7.21%) |
| CA-TCC | 44.75 | 52.09 | 45.63 | 46.26 | 71.33 | **100.0** | 52.71 | **99.85** | **99.84** | 46.48 | **100.0** | 77.01 | 69.66 |
| + Ours | **76.85** | **77.16** | **79.99** | **53.26** | **86.36** | **100.0** | **53.23** | 99.67 | 99.01 | **53.56** | **100.0** | **84.93** | **80.34** (+10.68%) |

(b) Transfer learning without adaptation under self- and semi-supervised settings on the FD datasets. TS-TCC and CA-TCC are used as baselines for self- and semi-supervised learning, respectively.

Table 4: Results of transfer learning task on both in- and cross-domain settings.

## 4.3 TRANSFER LEARNING

We conduct experiments on transfer learning for classification in in-domain and cross-domain settings which are used in previous works (Zhang et al., 2022; Eldele et al., 2021; 2023; Dong et al., 2023), by adopting our SoftCLT to TS-TCC and CA-TCC. As baseline methods, we consider TS-SD (Shi et al., 2021), TS2Vec (Yue et al., 2022), Mixing-Up (Wickstrøm et al., 2022), CLOCS (Kiyasseh et al., 2021), CoST (Woo et al., 2022), LaST (Wang et al., 2022), TF-C (Zhang et al., 2022), TS-TCC (Eldele et al., 2021), TST (Zerveas et al., 2021) and SimMTM (Dong et al., 2023). In in-domain transfer learning, the model is pretrained on SleepEEG (Kemp et al., 2000) and fine-tuned on Epilepsy (Andrzejak et al., 2001), where they are both EEG datasets and hence considered to be in a similar domain. In cross-domain transfer learning, which involves pretraining on one dataset and fine-tuning on different datasets, the model is pretrained on SleepEEG, and fine-tuned on three datasets from different domains, FD-B (Lessmeier et al., 2016), Gesture (Liu et al., 2009), and EMG (Goldberger et al., 2000). Also, we perform transfer learning without adaptation under self-and semi- supervised settings, where source and target datasets share the same set of classes but only 1% of labels are available for the source dataset, and no further training on the target dataset is allowed. Specifically, models are trained on one of the four conditions (A,B,C,D) in the Fault Diagnosis (FD) datasets (Lessmeier et al., 2016) and test on another. Table 4a shows the results of both in- and cross-domain transfer learning, and Table 4b shows the results of both self- and semi-supervised settings with FD datasets. Notably, SoftCLT applied to CA-TCC improves average accuracy of twelve transfer learning scenarios with FD datasets by 10.68%.

## 4.4 ANOMALY DETECTION

We conduct experiments on univariate TS anomaly detection (AD) task by adopting SoftCLT to TS2Vec (Yue et al., 2022) under two different settings: the normal setting splits each dataset into two halves according to the time order and use them for training and evaluation, respectively, and the cold-start setting pretrains models on the FordA dataset in the UCR archive and evaluates on each dataset. As baseline methods, we consider SPOT (Siffer et al., 2017), DSPOT (Siffer et al., 2017), DONUT (Xu et al., 2018), SR (Ren et al., 2019), for the normal setting, and FFT (Rasheed et al., 2009), Twitter-AD (Vallis et al., 2014), Luminol (LinkedIn, 2018) for the cold-start setting, and TS2Vec (Yue et al., 2022) for both. The anomaly score is computed by the L1 distance of two representations encoded from masked and unmasked inputs following TS2Vec. We evaluate the compared method on the Yahoo (Laptev et al., 2015) and KPI (Ren et al., 2019) datasets. We found that suppressing instance-wise CL leads to better AD performance on average, so we report TS2Vec and SoftCLT performances without instance-wise CL; more details can be found in the Appendix G. As shown in Table 5, SoftCLT outperforms the baselines in both settings in terms of the F1 score, precision, and recall. Specifically, SoftCLT applied to TS2Vec improves the F1 score approximately 2% in both datasets under both normal and cold-start settings.

| | Yahoo | | | KPI | | |
|---|---|---|---|---|---|---|
| | $F_1$ | Prec. | Rec. | $F_1$ | Prec. | Rec. |
| SPOT | 33.8 | 26.9 | 45.4 | 21.7 | 78.6 | 12.6 |
| DSPOT | 31.6 | 24.1 | 45.8 | 52.1 | 62.3 | 44.7 |
| DONUT | 2.6 | 1.3 | 82.5 | 34.7 | 37.1 | 32.6 |
| SR | 5.63 | 45.1 | 74.7 | 62.2 | 64.7 | 59.8 |
| TS2Vec* | 72.3 | 69.3 | 75.7 | 67.6 | 91.1 | 53.7 |
| + Ours | **74.2** | 72.2 | 76.5 | **70.1** | 91.6 | 57.0 |

(a) Results of AD task on normal setting.

| | Yahoo | | | KPI | | |
|---|---|---|---|---|---|---|
| | $F_1$ | Prec. | Rec. | $F_1$ | Prec. | Rec. |
| FFT | 29.1 | 20.2 | 51.7 | 53.8 | 47.8 | 61.5 |
| Twitter-AD | 24.5 | 16.6 | 46.2 | 33.0 | 41.1 | 27.6 |
| Luminol | 38.8 | 25.4 | 81.8 | 41.7 | 30.6 | 65.0 |
| SR | 52.9 | 40.4 | 76.5 | 66.6 | 63.7 | 69.7 |
| TS2Vec* | 74.0 | 70.7 | 77.6 | 68.9 | 89.3 | 56.2 |
| + Ours | **76.2** | 75.3 | 77.3 | **70.7** | 92.1 | 57.4 |

(b) Results of AD task on cold-start setting.

\* We used the official code to replicate the results without the instance-wise contrastive loss.

Table 5: Anomaly detection results.

| Soft assignment | | UCR datasets | UEA datasets |
|---|---|---|---|
| Instance-wise | Temporal | Avg. Acc.(%) | Avg. Acc.(%) |
| | | 82.3 | 70.5 |
| ✓ | | 83.9 (+1.6) | 73.0 (+2.5) |
| | ✓ | 83.7 (+1.4) | 73.8 (+3.3) |
| ✓ | ✓ | **85.0** (+2.7) | **74.2** (+3.7) |

(a) Application of soft assignments.

| Method | Avg. Acc.(%) |
|---|---|
| Neighbor | 76.1 |
| Linear | 77.2 |
| Gaussian | 83.5 |
| Sigmoid | **83.7** |

(b) Assignment func.

| $\alpha$ | Avg. Acc.(%) |
|---|---|
| 0.25 | 83.0 |
| 0.50 | **83.9** |
| 0.75 | 83.4 |
| 1.00 | 83.1 |

(c) Upper bound.

| Inst. CL | Temporal CL | |
|---|---|---|
| Metric | Hard | Soft |
| COS | 83.7 | 84.7 |
| EUC | 83.9 | 84.8 |
| DTW | 83.9 | **85.0** |
| TAM | 83.9 | **85.0** |

(d) Distance func.

Table 6: Ablation study results.

## 4.5 ABLATION STUDY

**Effectiveness of SoftCLT.** Table 6a shows the effect of soft assignments from the standard hard CL. Applying soft assignments to instance-wise or temporal CL provides a performance gain, and applying them to both dimensions results in the best performance, improving the accuracy on the UCR and UEA datasets by 2.7% and 3.7%, respectively.

**Design choices for soft temporal CL.** Table 6b compares different choices of the soft assignment $w_T$. **Neighbor** takes neighborhood within a window around the reference point as positive and the others as negative. **Linear** gives soft assignments linearly proportional to the time difference from the reference point, where the most distant one gets the value of zero. **Gaussian** gives soft assignments based on a Gaussian distribution with the mean of the reference point and the standard deviation as a hyperparameter. Among them, **Sigmoid** in Eq. 4 shows the best performance as shown in Table 6b.

**Upper bound for soft instance-wise CL.** In the soft instance-wise contrastive loss, $\alpha$ is introduced to avoid giving the same assignment to pairs of the same TS and pairs of the different TS with the distance of zero, where $\alpha = 1$ makes both cases to have the same assignment. Table 6c studies the effect of tuning $\alpha$. Based on the results, $\alpha = 0.5$ is the best, i.e., the similarity of the pairs of the same TS should be strictly larger than other pairs, but not by much.

**Distance metrics for soft instance-wise CL.** Table 6d compares different choices of the distance metric $D$ in Eq. 1: cosine distance (COS), Euclidean distance (EUC), dynamic time warping (DTW), and time alignment measurement (TAM) (Folgado et al., 2018) on 128 UCR datasets, where the baseline is TS2Vec and the hard or best soft temporal CL is applied together. The result shows that the improvement by soft instance-wise CL is robust to the choice of the distance metric. We use DTW throughout all other experiments because DTW is well-studied, commonly used in the literature and fast algorithms such as FastDTW are available.

## 4.6 ANALYSIS

**Comparison with soft CL methods in computer vision.** While soft CL methods have been proposed in other domains, they compute soft assignments on the embedding space because it is difficult to measure the similarities on the data space, particularly in computer vision. However, we argue that the similarities on the data space is indeed a strong self-supervision, leading to better representation learning. To confirm this, we compare SoftCLT with soft CL methods proposed in other domains working on the embedding space: NNCLR (Dwibedi et al., 2021) and ASCL (Feng & Patras, 2022), on UCR datasets. For a fair comparison, we apply all compared methods to TS2Vec under the same setting. As shown in Table 7, different from the proposed method, NNCLR and ASCL deteriorate the performance of TS2Vec, implying that similarities measured on the data space is strong self-supervision, while similarities measured on the learnable embedding space might not

| Method | Total | Length of time series | | Gap (A-B) |
|---|---|---|---|---|
| | | $\leq 200$ (A) | $> 200$ (B) | |
| TS2Vec | 82.3 | 88.1 | 79.6 | 5.8 |
| + NNCLR | 66.0 | 82.6 | 58.2 | 24.4 |
| + ASCL | 76.5 | 86.6 | 71.8 | 14.8 |
| + Ours | 85.0 | 89.8 | 81.9 | 7.9 |

| Temporal CL | Seasonality | |
|---|---|---|
| Soft | Low (103/128) | High (25/128) |
| ✗ | 84.1 | 80.1 |
| ✓ | 85.6 | 81.7 |
| Gain | +1.5 | +1.6 |

Table 7: Comparison of soft CL methods.     Table 8: Effect of soft temporal CL by seasonality.

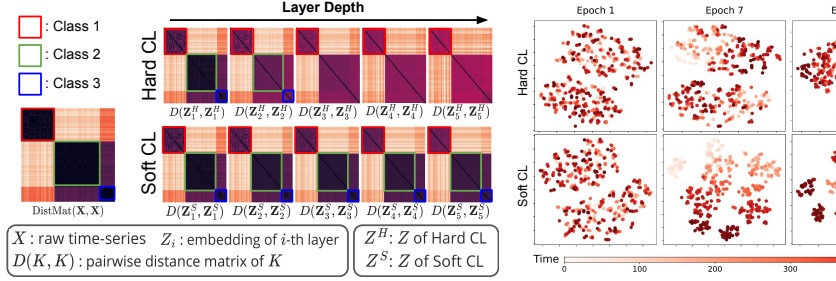

Figure 4: Pairwise distance matrices.     Figure 5: Visualization of temporal representations.

be useful in some domains. To further investigate the failure modes of the previous methods, we categorize datasets by the average TS length of 200 in Table 7, and observe that previous methods fail to capture the similarities of long TS data.

**Robustness to seasonality.** An assumption behind the proposed soft temporal CL is that values in adjacent timestamps are similar, which may raise a concern that seasonality in TS might not be captured. To address this, we categorize UCR datasets based on seasonality by ADF test (Sims et al., 1990) at the significance level of $p = 0.05$. As shown in Table 8, the performance gain by SoftCLT is consistent regardless of the seasonality. Our conjecture is that TS in the real world usually do not exhibit the perfect seasonality, as indicated by the ADF test result, such that SoftCLT takes advantage of the non-seasonal portions. Meanwhile, previous works have tried to decompose trend and seasonality in TS for representation learning (Wang et al., 2022; Woo et al., 2022). However, this may not be realistic for TS that are neither simultaneously auto-regressive nor stationary (Shen et al., 2022). In summary, we do not consider seasonality in TS directly, because it is not only challenging to extract but we can still achieve good performance without considering it in practice.

**Instance-wise relationships.** To see whether instance-wise relationships are preserved in the encoder, we visualize the pairwise instance-wise distance matrices of representations on the InsectEP-GRegularTrain dataset from the UCR archive (Dau et al., 2019) extracted from each layer, where the brighter color indicates the lower distance between instances. The top and bottom panels of Figure 4 show the changes in pairwise distance matrices of representations as depth progresses when adopting hard and soft CL, respectively. The results indicate that SoftCLT preserves the relationships between TS instances throughout encoding, while the standard hard CL fails to preserve them.

**Temporal relationships.** To assess the quality of temporal relationships captured by SoftCLT, we apply t-SNE (Van der Maaten & Hinton, 2008) to visualize the temporal representations, which are representations of each timestamp in a single TS. Figure 5 compares t-SNE of the representations learned with hard and soft CL over different training epochs, with the points getting darker as time progresses. While hard CL finds coarse-grained neighborhood relationships such that it fails to distinguish late timestamps in dark red, soft CL finds more fine-grained relationships.

## 5 CONCLUSION

In this paper, we present a soft contrastive learning framework for time series. In contrast to previous methods that give hard assignments to sample pairs, our approach gives soft assignments based on the instance-wise and temporal relationships on the data space. We demonstrate the effectiveness of our method in a range of tasks, leading to significant improvements in performance. We hope our work enlightens the effectiveness of self-supervision from the data space and motivates future works on contrastive representation learning in various domains to take account of it.

ETHICS STATEMENT

The proposed soft contrastive learning algorithm for time series has a potential to make a significant impact on the field of representation learning for time series data. The ability to apply this algorithm to various tasks and solve the general problem of time series representation learning is promising. In particular, the algorithm can be applied to transfer learning, which may be useful in scenarios with small datasets for downstream tasks. Furthermore, we expect that the idea of utilizing self-supervision from the data space for contrastive representation learning motivates future works in various domains.

However, as with any algorithm, there are ethical concerns to be considered. One potential ethical concern is a potential for the algorithm to perpetuate biases that may exist in the datasets used for pretraining. For example, if the pretraining dataset is imbalanced with respect to certain demographic attributes, this bias may be transferred to fine-tuning, potentially leading to biased predictions. It is essential to evaluate and address potential biases in the pretraining dataset before using the algorithm in real-world scenarios.

To ensure responsible use of the algorithm, we will make the datasets and code publicly available. Public availability of datasets and code allows for transparency and reproducibility, allowing other researchers to evaluate and address potential biases and misuse.

ACKNOWLEDGEMENTS

This work was supported by the National Research Foundation of Korea (NRF) grant funded by the Korea government (MSIT) (2020R1A2C1A01005949, 2022R1A4A1033384, RS-2023-00217705), the MSIT(Ministry of Science and ICT), Korea, under the ICAN(ICT Challenge and Advanced Network of HRD) support program (RS-2023-00259934) supervised by the IITP(Institute for Information & Communications Technology Planning & Evaluation), the Yonsei University Research Fund (2023-22-0071), and the Son Jiho Research Grant of Yonsei University (2023-22-0006).

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

# A  Dataset Description

## A.1  Classification

For time series classification, we use the UCR archive (Dau et al., 2019) and UEA archive (Bagnall et al., 2018). The UCR archive contains 128 univariate datasets, while the UEA archive contains 30 multivariate datasets. Among them, some datasets cannot be handled by T-Loss (Franceschi et al., 2019), TS-TCC (Eldele et al., 2021), and TNC (Tonekaboni et al., 2021) due to missing observations, such as DodgerLoopDay, DodgerLoopGame, and DodgerLoopWeekend. Additionally, there is no reported result for the DTW-D (Chen et al., 2013) on the InsectWingbeat dataset in the UEA archive. Hence, the comparison is conducted using the remaining 125 UCR datasets and 29 UEA datasets in the main paper. However, TS2Vec works well on all UCR and UEA datasets, so we experiment with all 128 UR datasets and 30 UEA datasets for ablation studies for our method on top of TS2Vec.

## A.2  Semi-supervised Classification

Table A.1 describes the summary of the statistical information for eight datasets (Anguita et al., 2013; Andrzejak et al., 2001; Dau et al., 2019) used in semi-supervised classifiaction, including the number of training and testing samples, data length, the number of sensor channels, and the number of classes.

| Dataset | # Train | # Test | Length | # Channel | # Class |
|---|---|---|---|---|---|
| HAR | 7,352 | 2,947 | 128 | 9 | 6 |
| Epilepsy | 9,200 | 2,300 | 178 | 1 | 2 |
| Wafer | 1,000 | 6,174 | 152 | 1 | 2 |
| FordA | 1,320 | 3,601 | 500 | 1 | 2 |
| FordB | 3,636 | 810 | 500 | 1 | 2 |
| POC | 1,800 | 858 | 80 | 1 | 2 |
| StarLightCurves | 1,000 | 8,236 | 1,024 | 1 | 3 |
| ElectricDevices | 8,926 | 7,711 | 96 | 1 | 7 |

Table A.1: Eight datasets used for semi-supervised classification

## A.3  Transfer Learning

We evaluate our approach on various datasets, which cover a wide range of application scenarios, including neurological healthcare, human activity recognition, mechanical fault detection, and physical status monitoring. Table A.2 describes the datasets for in-domain and cross-domain transfer learning. Fault Diagnosis (FD) datasets were used for transfer learning under self- and semi-supervised settings. The data statistics are described below.

| | | Dataset | # Samples | # Channels | # Classes | Length | Freq (Hz) |
|---|---|---|---|---|---|---|---|
| Pre-training | | SleepEEG | 371,055 | 1 | 5 | 200 | 100 |
| Fine-tuning | In-domain | Epilepsy | 60 / 20 / 11,420 | 1 | 2 | 178 | 174 |
| | Cross-domain | FD-B | 60 / 21 / 13,559 | 1 | 3 | 5,120 | 64,000 |
| | | Gesture | 320 / 120 / 120 | 3 | 8 | 315 | 100 |
| | | EMG | 122 / 41 / 41 | 1 | 3 | 1,500 | 4,000 |

Table A.2: In the four application scenarios, we utilize a pre-training dataset and a fine-tuning dataset, with the latter having a sample size denoted by "A/B/C," where each denotes the number of samples used for fine-tuning, validation, and testing, respectively. Our evaluation also focuses on small datasets, with a very limited sample size of less than 320 samples in the fine-tuning dataset, ensuring that the fine-tuning set is balanced in terms of classes. This approach enables us to test our model's effectiveness on small datasets, which has practical significance.

(1) **SleepEEG** (Kemp et al., 2000) dataset contains EEG recordings of 153 whole-night sleep sessions from 82 healthy individuals. We segmented the EEG signals using a non-overlapping approach, following the same preprocessing method as (Zhang et al., 2022), to obtain 371,055 univariate brainwaves, each sampled at 100 Hz and categorized into one of five sleep stages: Wake, Non-rapid eye movement (3 sub-states), and Rapid Eye Movement. When using SleepEEG dataset as a source

dataset in transfer learning task, we use cosine similarity instead of DTW due to the property of EEG datasets (Li et al., 2022).

(2) **Epilepsy** (Andrzejak et al., 2001) dataset monitors brain activity using a single-channel EEG sensor on 500 subjects, with each subject being recorded for 23.6 seconds. The dataset is sampled at 178 Hz and contains 11,500 samples. We followed the same preprocessing method as (Zhang et al., 2022) and classified the first four classes (eyes open, eyes closed, EEG measured in the healthy brain region, and EEG measured in the tumor region) of each sample as positive, while the remaining classes (whether the subject has a seizure episode) were classified as negative.

(3) **FD-B** (Lessmeier et al., 2016) dataset is collected from electromechanical drive systems and monitors the condition of rolling bearings to detect their failures based on monitoring conditions such as speed, load torque, and radial force. It consists of 13,640 samples, each recorded at 64k Hz and categorized into three classes: undamaged, inner damaged, and outer damaged.

(4) **Gesture** (Liu et al., 2009) dataset includes data on 8 hand gestures based on hand movement paths recorded by an accelerometer. The eight gestures are hand swiping left, right, up, and down, hand waving in a counterclockwise or clockwise circle, hand waving in a square, and waving a right arrow. The dataset contains 440 balanced classification labels, with each sample having eight different categories of gestures.

(5) **EMG** (Goldberger et al., 2000) dataset consists of 163 single-channel EMG recordings from the tibialis anterior muscle of three healthy volunteers suffering from neuropathy and myopathy. Each sample is associated with one of three classes, with each class representing a different patient. The dataset is sampled at 4K Hz.

(6) **FD** (Lessmeier et al., 2016) dataset was obtained by monitoring the sensor readings of a bearing machine while it operated under four distinct working conditions. Each working condition can be regarded as a separate domain since they exhibit unique features, such as variations in rotational speed and load torque. Within each domain, there are three categories: two fault classes, inner and outer fault, and one healthy class. The FD dataset has 8,184 training samples, 2,728 test samples, a data length of 5,120, one channel, and three classes. Our main goal is to use this dataset to conduct transferability experiments under both self- and semi-supervised settings and demonstrate the efficiency of our approach in transfer learning situations.

### A.4 ANOMALY DETECTION

We employed Yahoo (Laptev et al., 2015) and KPI (Ren et al., 2019) for the anomaly detection task. Yahoo is a benchmark dataset that contains 367 hourly sampled time series with annotated anomaly points. This dataset covers a wide range of anomaly types, including outliers and change-points. KPI is a competition dataset released by AIOPS Challenge in 2019. It contains several minutely sampled real KPI curves from diverse internet companies.

## B BASELINE METHODS

**Classification:** The results of all baseline methods for the classification task (DTW-D (Chen et al., 2013), TNC (Tonekaboni et al., 2021), TST (Zerveas et al., 2021), TS-TCC (Eldele et al., 2021), T-Loss (Franceschi et al., 2019), and TS2Vec (Yue et al., 2022)) are reported in Yue et al. (2022).

- DTW-D (Chen et al., 2013): DTW-D (Dynamic Time Warping-Delta) is a variant of DTW under semi-supervised learning settings.

- TNC (Tonekaboni et al., 2021): TNC (Temporal Neighborhood Coding) defines temporal neighborhood of window using normal distribution, and defines samples in neighborhood and non-neighborhood as positives and negatives, respectively.

- TST (Zerveas et al., 2021): TST (Time Series Transformer) adopts the masked modeling paradigm to time series domain, where the goal is to reconstruct the masked time stamps.

- TS-TCC (Eldele et al., 2021): TS-TCC (Time-Series representation learning framework via Temporal and Contextual Contrasting) proposes a new temporal contrastive loss by making the augmentations predict each other's future.

- T-Loss (Franceschi et al., 2019): T-Loss is a triplet loss designed for time series. It samples a random subseries from a time series and treats them as positive when they belong to its subseries, and negative if belong to subseries of other time series.

- TS2Vec (Yue et al., 2022): TS2Vec splits time series into several subseries and defines hierarchical contrastive loss in both instance-wise and temporal dimensions.

**Semi-supervised classification:** The results of all baseline methods for semi-supervised classification using self-supervised methods (SSL-ECG (Sarkar & Etemad, 2020), CPC (Oord et al., 2018), SimCLR (Chen et al., 2020), TS-TCC (Eldele et al., 2021)) and semi-supervised methods (Mean-Teacher (Tarvainen & Valpola, 2017), DivideMix (Li et al., 2020), SemiTime (Fan et al., 2021), FixMatch (Sohn et al., 2020), CA-TCC (Eldele et al., 2023)) are reported in Eldele et al. (2023).

- SSL-ECG (Sarkar & Etemad, 2020): SSL-ECG (Self-supervised ECG Representation Learning for Emotion Recognition) proposes ECG-based emotion recognition using multi-task self-supervised learning

- CPC (Oord et al., 2018): CPC (Contrastive Predictive Coding) combines predicting future observations (predictive coding) with a probabilistic contrastive loss.

- SimCLR (Chen et al., 2020): SimCLR proposes a simple framework for contrastive learning of visual representations, without requiring specialized architectures or a memory bank.

- Mean-Teacher (Tarvainen & Valpola, 2017): Mean-Teacher is an algorithm for semi-supervised algorithm, that averages model weights instead of predictions.

- DivideMix (Li et al., 2020): DivideMix uses a mixture model to divide training data into labeled clean samples and unlabeled noisy samples, and trains a model on both sets in a semi-supervised way.

- SemiTime (Fan et al., 2021): SemiTime conducts supervised classification on labeled time series data and self-supervised prediction of temporal relations on unlabeled time series data. It achieves this by sampling segments of past-future pairs from the same or different candidates and training the model to distinguish between positive and negative temporal relations between those segments.

- FixMatch (Sohn et al., 2020): FixMatch generates pseudo-labels using the model's predictions on weakly-augmented unlabeled images, and retain the pseudo-label with a high-confidence prediction. Then, the model is trained to predict the pseudo-label when fed a strongly-augmented version of the same image.

- CA-TCC (Eldele et al., 2023): CA-TCC (Self-supervised Contrastive Representation Learning for Semi-supervised Time-Series Classification) is the extension of TS-TCC to the semi-supervised settings, and adopts the same contrastive loss as TS-TCC.

**Transfer learning:** The results of baseline methods for transfer learning in both in-domain and cross-domain settings (TS-SD (Shi et al., 2021), TS2Vec (Yue et al., 2022), Mixing-Up (Wickstrøm et al., 2022), TF-C (Zhang et al., 2022), TS-TCC (Eldele et al., 2021), TST (Zerveas et al., 2021), SimMTM (Dong et al., 2023)) using SleepEEG dataset as the pre-training dataset, are reported in Dong et al. (2023), except for results of TS-SD which are reported in Zhang et al. (2022). The results of baseline methods for transfer learning in both self-supervised and semi-supervised settings (Supervised, TS-TCC (Eldele et al., 2021), CA-TCC (Eldele et al., 2023)), using FD dataset as the pre-training dataset, are reported in Eldele et al. (2023).

- TS-SD (Shi et al., 2021): TS-SD utilizes a triplet similarity discrimination task to train a model. The objective is to determine which of the two TS is more similar to a given TS, with DTW employed as a means to define the similarity.

- Mixing-Up (Wickstrøm et al., 2022): Mixing-up generates new time series by mixing two time series, and predicts the mixing weights.

- TF-C (Zhang et al., 2022): TF-C generates both time-based and frequency-based representations of time series and proposes a novel time-frequency consistency architecture.

- SimMTM (Dong et al., 2023): SimMTM adopts the masked modeling paradigm to time series domain, where the goal is to reconstruct the original time series from multiple masked series.

**Anomaly detection:** The results of all baseline methods for the anomaly detection task (SPOT (Siffer et al., 2017), DSPOT (Siffer et al., 2017), DONUT (Xu et al., 2018), SR (Ren et al., 2019), FFT (Rasheed et al., 2009), Twitter-AD (Vallis et al., 2014), Luminol (LinkedIn, 2018), TS2Vec (Yue et al., 2022)) are reported in Yue et al. (2022).

- SPOT (Siffer et al., 2017): SPOT is a novel outlier detection approach for streaming univariate time series, based on Extreme Value Theory, which does not rely on pre-set thresholds, assumes no distribution, and only requires a single parameter to control the number of false positives.

- DONUT (Xu et al., 2018): DONUT is an unsupervised anomaly detection algorithm based on variational autoencoder.

- SR (Ren et al., 2019): SR is a time-series anomaly detection algorithm that is based on the Spectral Residual (SR) model and Convolutional Neural Network (CNN), where the SR model is borrowed from visual saliency detection and combined with CNN to improve its performance.

- FFT (Rasheed et al., 2009): FFT uses fast fourier transform to detect the areas with high frequency change.

- Twitter-AD (Vallis et al., 2014): Twitter-AD automatically detects long-term anomalies in cloud data by identifying anomalies in application and system metrics.

- Luminol (LinkedIn, 2018): Luminol is a Python library for time series data analysis that provides two main functionalities - anomaly detection and correlation - and can be utilized to investigate the potential causes of anomalies.

## C   IMPLEMENTATION DETAILS

The table of hyperparameter settings that we utilized can be found in Table C.1. We made use of five hyperparameters: $\tau_I$, $\tau_T$, $\lambda$, batch size (bs), and learning rate (lr). For semi-supervised classification and transfer learning, we set the weight decay to 3e-4, $\beta_1 = 0.9$, and $\beta_2 = 0$. The number of optimization iterations for classification and anomaly detection tasks is set to 200 for datasets with a size less than 100,000; otherwise, it is set to 600. Additionally, the training epochs for semi-supervised classification are set to 80, while for transfer learning, it is set to 40.

Since we utilized soft contrastive loss as an auxiliary loss for TS-TCC and CA-TCC, which are the methods involved in solving semi-supervised classification and transfer learning tasks, we introduced an additional hyperparameter $\lambda_{aux}$ to control the contribution of the auxiliary loss to the final loss.

|  | Classification / Forecasting | Semi-supervised classification | Transfer learning | Anomaly detection |
|---|---|---|---|---|
| $\tau_I$ | [1, 2, 3, 4, 5, 10, 20] | [10, 20, 30, 40, 50] | | |
| $\tau_T$ | [0.5, 1.0, 1.5, 2.0, 2.5] | [1.5, 2.0, 2.5] | | |
| $\lambda$ | 0.5 | [0.3, 0.5] | | 0.5 |
| $\lambda_{aux}$ | - | [0.1, 0.3, 0.5] | | - |
| bs | 8 | 16 | | 4 (yahoo) / 8 (kpi) |
| lr | 0.001 | 0.0003 | | 0.001 |

Table C.1: Hyperparameter settings for various tasks

## D   PROBABILISTIC INTERPRETATION OF SOFT CONTRASTIVE LOSSES

Inspired by the fact that the contrastive loss can be interpreted as the cross-entropy loss with virtual labels defined per batch, or equivalently, the KL divergence of the predicted softmax probability from the virtual label or hard assignment (Lee et al., 2021), we define a softmax probability of the relative similarity out of all similarities considered when computing the loss, and interpret our soft contrastive losses as a weighted sum of the cross-entropy losses. In this section, we show that the proposed contrastive loss can also be seen as the scaled KL divergence of the predicted softmax probabilities from the normalized soft assignments, where the scale is the sum of soft assignments. When hard assignment is applied, the loss becomes the standard contrastive loss, which is often called InfoNCE (Oord et al., 2018).

### D.1 Probabilistic Interpretation of Soft Instance-Wise Contrastive Loss

To simplify indexing, we extend soft assignments to incorporate the positive sample and anchor itself:

$$w'_I(i, i') = \begin{cases} 0, & \text{if } i = i'; \\ 1, & \text{if } i \neq i' \text{ and } i \equiv i' (\text{mod } N); \\ w_I(i, i' \text{ mod } N), & \text{otherwise}; \end{cases} \tag{D.1}$$

and let $q_I(i, i') = w'_I(i, i')/Z_I$ be its normalization, where $Z_I = \sum_{j=1}^{2N} w'_I(i, j)$ is the partition function. Then, we can rewrite the proposed soft instance-wise contrastive loss as follows:

$$\ell_I^{(i,t)} = -\log p_I((i, i+N), t) - \sum_{j=1, j \neq \{i, i+N\}}^{2N} w_I(i, j \text{ mod } N) \cdot \log p_I((i, j), t)$$

$$= -\sum_{j=1}^{2N} w'_I(i, j) \cdot \log p_I((i, j), t)$$

$$= -Z_I \cdot \sum_{j=1}^{2N} \frac{w'_I(i, j)}{Z_I} \cdot \log p_I((i, j), t)$$

$$= Z_I \cdot \sum_{j=1}^{2N} q_I(i, j) \cdot \log \frac{q_I(i, j)}{p_I((i, j), t)} - \underbrace{q_I(i, j) \log q_I(i, j)}_{= \text{ constant}}. \tag{D.2}$$

Let $Q_I$ and $P_I$ be the probability distributions of $q_I(i, j)$, and $p_I((i, j), t)$, respectively. Then, we can rewrite the above loss as:

$$\ell_I^{(i,t)} = Z_I \cdot KL(Q_I || P_I) + \text{const}, \tag{D.3}$$

which is the scaled KL divergence of the predicted softmax probability from the soft assignments.

### D.2 Probabilistic Interpretation of Soft Temporal Contrastive Loss

To simplify indexing, we extend soft assignments to incorporate the positive sample and anchor itself:

$$w'_T(t, t') = \begin{cases} 0, & \text{if } t = t'; \\ 1, & \text{if } t \neq t' \text{ and } t \equiv t' (\text{mod } T); \\ w_T(t, t' \text{ mod } T), & \text{otherwise}; \end{cases} \tag{D.4}$$

and let $q_T(t, t') = w'_T(t, t')/Z_T$ be its normalization, where $Z_T = \sum_{s=1}^{2T} w'_T(t, s)$ is the partition function. Then, we can rewrite the proposed soft temporal contrastive loss as follows:

$$\ell_T^{(i,t)} = -\log p_T(i, (t, t+T)) - \sum_{s=1, s \neq \{t, t+T\}}^{2T} w_T(t, s \text{ mod } N) \cdot \log p_T(i, (t, s))$$

$$= -\sum_{s=1}^{2T} w'_T(t, s) \cdot \log p_T(i, (t, s))$$

$$= -Z_T \cdot \sum_{s=1}^{2T} \frac{w'_T(t, s)}{Z_T} \cdot \log p_T(i, (t, s))$$

$$= Z_T \cdot \sum_{s=1}^{2T} q_T(t, s) \cdot \log \frac{q_T(t, s)}{p_T(i, (t, s))} - \underbrace{q_T(t, s) \log q_T(t, s)}_{= \text{ constant}}. \tag{D.5}$$

Let $Q_T$ and $P_T$ be the probability distributions of $q_T(t, s)$, and $p_T(i, (t, s))$, respectively. Then, we can rewrite the above loss as:

$$\ell_T^{(i,t)} = Z_T \cdot KL(Q_T || P_T) + \text{const}, \tag{D.6}$$

which is the scaled KL divergence of the predicted softmax probability from the soft assignments. These answer to a concern that targets are fixed while the predicted softmax probabilities are relative to the samples in the batch: the formulation with fixed targets is proportional to the formulation with relative targets, and their difference is only in the optimization speed by the scale $Z_I$ and $Z_T$.

## E    HIERARCHICAL SOFT TEMPORAL CONTRASTIVE LOSS

In our approach, we adopt the hierarchical contrastive loss proposed in TS2Vec (Yue et al., 2022), where we apply max-pooling on the representations along the temporal axis and contrastive learning is performed at each level. However, as max pooling proceeds, semantic similarities between adjacent time step decrease, so the sharpness needs to be adjusted based on the hierarchy depth and kernel size.

| Sharpness | Avg. Acc.(%) |
|---|---|
| $\tau_T$ | 83.3 |
| $m^k \cdot \tau_T$ | **83.7** |

Table E.1: Effect of hierarchical $\tau_T$

To address this, we increase the value of sharpness in soft temporal contrastive loss as the depth of the network increases, thereby reflecting the hierarchy of the time series. That is, we use $m^k \cdot \tau_T$ instead of $\tau_T$ for the sharpness value in soft temporal contrastive loss, where $m$ is the kernel size of pooling layers and $k$ is the depth. For all datasets, we set $m$ to 2, while the value of $k$ depends on the length of each specific dataset. We conducted an ablation study to assess the effect of using hierarchical sharpness, by comparing the performance of using hierarchical sharpness ($m^k \cdot \tau_T$) against a constant sharpness ($\tau_T$) using 128 datasets in UCR archive (Dau et al., 2019). To solely observe the effect of hierarchical temporal contrastive loss, we employ the original hard instance-wise contrastive loss for this experiment. The results presented in Table E.1 demonstrate that increasing $\tau_T$ as the depth of the network increases leads to improved performance.

## F    DESIGN FOR INSTANCE-WISE CONTRASTIVE LOSS

In this study, we explore different options for the soft assignments used in the soft instance-wise contrastive loss: without kernel (*w/o kernel*), Laplacian kernel, and Gaussian kernel. For *w/o kernel*, we use $w_I(i,j) = 1 - D(x_i, x_j)$, where $D$ is a min-max normalized distance metric. For the Laplacian kernel, we use $w_I(i,j) = \exp\left(-\frac{D(x_i,x_j)}{\sigma}\right)$, and for the Gaussian kernel, we use $w_I(i,j) = \exp\left(-\frac{(D(x_i,x_j))^2}{2\sigma^2}\right)$, where $\sigma$ is a hyperparameter. For all kernels, sample pairs with a lower distance

| Method | Avg. Acc.(%) |
|---|---|
| w/o kernel | 79.1 |
| Gaussian | 82.6 |
| Laplacian | 83.1 |
| Sigmoid | **83.9** |

Table F.1: Design for instance-wise CL

tends to have soft assignments closer to one. We conduct an ablation study by comparing the performance of using the above kernels to model the soft assignments using the UCR archive datasets, and the results are presented in Table F.1. For this experiment, we employ the original hard temporal contrastive loss to solely observe the effect of the functions used for the instance-wise contrastive loss.

## G    CONTRASTIVE LEARNING FOR ANOMALY DETECTION TASK

Table G.1 indicates that employing only temporal contrastive loss, while excluding instance-wise contrastive loss, yields better performance in the majority of hard CL and soft CL settings for anomaly detection tasks. This can be attributed to the nature of the anomaly detection task, which involves detecting anomalies within a time series, and is less concerned with other time series.

| | | Yahoo | | | KPI | | | | | Yahoo | | | KPI | | |
|---|---|---|---|---|---|---|---|---|---|---|---|---|---|---|---|
| | | $F_1$ | Prec. | Rec. | $F_1$ | Prec. | Rec. | | | $F_1$ | Prec. | Rec. | $F_1$ | Prec. | Rec. |
| TS2Vec | w/ inst | **72.4** | **69.3** | 75.7 | 67.6 | 90.9 | 53.7 | TS2Vec | w/ inst | 74.0 | 70.7 | **77.6** | 68.9 | **89.3** | 56.2 |
| | w/o inst | 71.8 | 67.6 | **76.5** | **68.3** | **90.9** | **54.6** | | w/o inst | **75.5** | **73.6** | 77.4 | **69.7** | 88.8 | **57.4** |
| + Ours | w/ inst | 71.2 | 67.8 | 74.9 | 66.4 | **94.3** | 51.4 | + Ours | w/ inst | 74.6 | 72.1 | **77.5** | 69.0 | **92.1** | 56.2 |
| | w/o inst | **74.2** | **72.2** | **76.5** | **70.1** | 91.6 | **57.0** | | w/o inst | **76.2** | **75.3** | 77.3 | **69.7** | **92.1** | **57.4** |

(a) Results of AD task on normal setting                    (b) Results of AD task on cold-start setting

Table G.1: Results of anomaly detection task by the use of instance-wise contrastive loss.

# H  TIME SERIES FORECASTING

The tasks mentioned in the main paper, except for anomaly detection, can be classified as high-level tasks, which requires capturing instance-wise representations. High-level tasks generally perform better with CL methods than with masked modeling methods (Dong et al., 2023; Huang et al., 2022; Xie et al., 2022). However, we can perform low-level tasks such as time series forecasting, when using encoder architectures that can obtain representations of each timestamp.

For TS forecasting, we apply SoftCLT to both TS2Vec and CoST (Woo et al., 2022). Capturing temporal information within time series is crucial for time series forecasting, so we use soft CL in two ways for TS2Vec: by adopting only temporal contrastive loss and by using both temporal and instance-wise contrastive loss. For the experiment, we use four datasets, ETTh1, ETTh2, ETTm1, and electricity dataset (Zhou et al., 2021) for TS2Vec, and ETTh1, ETTh2, ETTm1, and Weather dataset (Zhou et al., 2021) for CoST, under both univariate and multivariate settings. Table H.1 describes the summary of the statistical information for the five datasets. As demonstrated in Table H.2 and Table H.3, our method results in performance gains compared to hard CL in both univariate and multivariate TS forecasting for both TS2Vec and CoST.

| Datasets | Channels | Prediction Length | Samples |
|---|---|---|---|
| ETTh$_1$, ETTh$_2$ Electricity Weather | 7 321 21 | {24,48,168,336,720} | 8640 / 2880 / 2880 15782 / 5261 / 5261 36792 / 5271 / 10540 |
| ETTm$_1$ | 7 | {24,48,96,288,672} | 34560 / 11520 / 11520 |

Table H.1: Four datasets used for time series forecasting, organized in the format of train/valid/test.

| | | Univariate forecasting | | | | | | | | Multivariate forecasting | | | | | | | |
| | | w/ instance-wise CL | | | | w/o instance-wise CL | | | | w/ instance-wise CL | | | | w/o instance-wise CL | | | |
| | | TS2Vec | | + Ours | | TS2Vec | | + Ours | | TS2Vec | | + Ours | | TS2Vec | | + Ours | |
| Dataset | H | MSE | MAE | MSE | MAE | MSE | MAE | MSE | MAE | MSE | MAE | MSE | MAE | MSE | MAE | MSE | MAE |
|---|---|---|---|---|---|---|---|---|---|---|---|---|---|---|---|---|---|
| ETTh$_1$ | 24 | 0.042 | **0.152** | **0.041** | 0.156 | 0.046 | 0.164 | **0.045** | **0.161** | 0.568 | 0.513 | **0.554** | **0.506** | 0.568 | 0.525 | **0.554** | **0.510** |
| | 48 | 0.067 | 0.197 | **0.064** | **0.194** | **0.079** | **0.216** | 0.080 | 0.218 | 0.607 | 0.538 | **0.595** | **0.532** | 0.617 | 0.557 | **0.593** | **0.542** |
| | 168 | 0.154 | 0.304 | **0.144** | **0.293** | 0.153 | 0.302 | **0.144** | **0.291** | 0.742 | **0.622** | **0.737** | 0.624 | 0.796 | 0.664 | **0.765** | **0.647** |
| | 336 | 0.174 | 0.332 | **0.162** | **0.318** | 0.172 | 0.328 | **0.160** | **0.314** | 0.937 | 0.726 | **0.890** | **0.712** | 1.024 | 0.777 | **0.867** | **0.702** |
| | 720 | 0.209 | 0.376 | **0.179** | **0.345** | 0.192 | 0.357 | **0.178** | **0.341** | 1.068 | 0.800 | **1.056** | **0.798** | 1.063 | 0.801 | **1.046** | **0.795** |
| | Avg. | 0.129 | 0.272 | **0.120** | **0.261** | 0.128 | 0.273 | **0.121** | **0.265** | 0.784 | 0.640 | **0.766** | **0.634** | 0.814 | 0.665 | **0.765** | **0.639** |
| ETTh$_2$ | 24 | 0.090 | 0.230 | **0.086** | **0.224** | 0.090 | 0.229 | **0.088** | **0.226** | 0.373 | 0.465 | **0.370** | **0.462** | 0.371 | 0.462 | **0.362** | **0.452** |
| | 48 | 0.126 | 0.273 | **0.121** | **0.268** | 0.121 | 0.268 | **0.119** | **0.265** | 0.561 | 0.579 | **0.557** | **0.577** | 0.548 | 0.571 | **0.535** | **0.559** |
| | 168 | 0.208 | 0.359 | **0.202** | **0.354** | 0.196 | 0.349 | **0.194** | **0.347** | **1.713** | **1.015** | **1.713** | 1.016 | 1.693 | 1.024 | **1.606** | **1.001** |
| | 336 | 0.219 | 0.374 | **0.206** | **0.363** | 0.207 | 0.364 | **0.205** | **0.362** | 2.153 | 1.167 | **2.061** | **1.147** | 2.096 | 1.172 | **1.973** | **1.135** |
| | 720 | 0.221 | 0.381 | **0.216** | **0.377** | 0.217 | 0.377 | **0.215** | **0.376** | 2.437 | 1.299 | **2.394** | **1.275** | 2.464 | 1.319 | **2.297** | **1.259** |
| | Avg. | 0.173 | 0.323 | **0.166** | **0.316** | 0.166 | 0.317 | **0.164** | **0.315** | 1.447 | 0.905 | **1.441** | **0.895** | 1.434 | 0.910 | **1.355** | **0.881** |
| ETTm$_1$ | 24 | 0.016 | 0.093 | **0.014** | **0.088** | 0.015 | 0.092 | **0.014** | **0.090** | 0.459 | 0.449 | **0.418** | **0.426** | 0.428 | 0.430 | **0.421** | **0.423** |
| | 48 | 0.029 | 0.128 | **0.027** | **0.124** | 0.028 | 0.126 | **0.027** | **0.124** | 0.608 | 0.521 | **0.567** | **0.501** | 0.587 | 0.512 | **0.568** | **0.501** |
| | 96 | 0.044 | 0.158 | **0.041** | **0.155** | **0.048** | **0.166** | 0.048 | 0.166 | 0.597 | 0.532 | **0.591** | **0.530** | 0.623 | 0.544 | **0.595** | **0.524** |
| | 288 | 0.103 | 0.246 | **0.093** | **0.232** | **0.113** | **0.258** | 0.115 | 0.260 | 0.670 | 0.586 | **0.647** | **0.577** | 0.704 | 0.600 | **0.659** | **0.580** |
| | 672 | 0.155 | 0.298 | **0.135** | **0.283** | 0.163 | 0.313 | **0.160** | **0.311** | 0.750 | 0.639 | **0.743** | **0.637** | 0.797 | 0.659 | **0.753** | **0.642** |
| | Avg. | 0.069 | 0.185 | **0.062** | **0.176** | **0.073** | 0.191 | **0.073** | **0.190** | 0.617 | 0.545 | **0.593** | **0.534** | 0.628 | 0.549 | **0.599** | **0.534** |
| Electricity | 24 | 0.259 | 0.291 | **0.251** | **0.284** | 0.268 | 0.299 | **0.252** | **0.286** | **0.285** | 0.375 | 0.286 | **0.375** | 0.317 | 0.400 | **0.315** | **0.398** |
| | 48 | 0.309 | 0.323 | **0.304** | **0.317** | 0.326 | 0.350 | **0.306** | **0.323** | **0.308** | 0.391 | **0.308** | **0.391** | 0.340 | 0.415 | **0.338** | **0.413** |
| | 168 | 0.426 | 0.397 | **0.418** | **0.391** | 0.446 | 0.431 | **0.425** | **0.401** | 0.335 | 0.411 | **0.334** | **0.411** | 0.364 | 0.432 | **0.362** | **0.430** |
| | 336 | 0.567 | 0.484 | **0.560** | **0.479** | 0.589 | 0.524 | **0.571** | **0.494** | 0.352 | 0.424 | **0.351** | **0.424** | 0.380 | 0.443 | **0.377** | **0.441** |
| | 720 | 0.860 | 0.650 | **0.858** | **0.645** | 0.882 | 0.700 | **0.879** | **0.685** | **0.378** | 0.442 | **0.378** | **0.442** | 0.403 | 0.459 | **0.401** | **0.457** |
| | Avg. | 0.484 | 0.429 | **0.478** | **0.423** | 0.502 | 0.461 | **0.451** | **0.438** | 0.332 | **0.409** | **0.331** | **0.409** | 0.361 | 0.430 | **0.359** | **0.428** |

Table H.2: Results of univariate and multivariate time series forecasting.

| Dataset | H | Univariate forecasting | | | | Multivariate forecasting | | | |
| | | CoST | | + Ours | | CoST | | + Ours | |
| | | MSE | MAE | MSE | MAE | MSE | MAE | MSE | MAE |
|---|---|---|---|---|---|---|---|---|---|
| ETTh$_1$ | 24 | **0.040** | **0.152** | **0.040** | **0.152** | 0.386 | 0.429 | **0.377** | **0.422** |
| | 48 | 0.064 | **0.186** | **0.061** | **0.186** | 0.437 | 0.464 | **0.428** | **0.455** |
| | 168 | 0.097 | 0.236 | **0.093** | **0.230** | 0.643 | 0.582 | **0.626** | **0.571** |
| | 336 | 0.112 | 0.258 | **0.110** | **0.254** | 0.812 | 0.679 | **0.773** | **0.660** |
| | 720 | 0.158 | 0.316 | **0.155** | **0.314** | 0.970 | 0.771 | **0.891** | **0.744** |
| | Avg. | 0.094 | 0.230 | **0.091** | **0.227** | 0.650 | 0.585 | **0.619** | **0.570** |
| ETTh$_2$ | 24 | 0.079 | 0.207 | **0.078** | **0.206** | 0.480 | 0.525 | **0.472** | **0.522** |
| | 48 | 0.118 | 0.259 | **0.117** | **0.257** | 0.751 | 0.669 | **0.749** | **0.667** |
| | 168 | 0.189 | 0.339 | **0.179** | **0.332** | 1.613 | 1.017 | **1.602** | **1.009** |
| | 336 | 0.206 | 0.360 | **0.201** | **0.357** | 1.807 | 1.078 | **1.800** | **1.075** |
| | 720 | 0.214 | 0.371 | **0.208** | **0.369** | 1.959 | 1.099 | **1.951** | **1.090** |
| | Avg. | 0.161 | 0.307 | **0.156** | **0.304** | 1.322 | 0.876 | **1.315** | **0.872** |
| ETTm$_1$ | 24 | 0.015 | 0.088 | **0.014** | **0.087** | 0.246 | 0.329 | **0.243** | **0.326** |
| | 48 | 0.025 | 0.117 | **0.024** | **0.116** | **0.331** | **0.386** | 0.332 | **0.386** |
| | 96 | 0.038 | 0.147 | **0.036** | **0.145** | 0.378 | 0.419 | **0.376** | **0.418** |
| | 288 | 0.077 | 0.209 | **0.075** | **0.204** | **0.472** | 0.486 | 0.474 | **0.484** |
| | 672 | 0.113 | 0.257 | **0.105** | **0.245** | 0.620 | 0.574 | **0.613** | **0.567** |
| | Avg. | 0.054 | 0.164 | **0.051** | **0.159** | 0.409 | 0.439 | **0.407** | **0.436** |
| Weather | 24 | 0.096 | 0.213 | **0.095** | **0.212** | 0.299 | 0.361 | **0.298** | **0.360** |
| | 48 | 0.139 | 0.263 | **0.132** | **0.260** | 0.359 | 0.411 | **0.357** | **0.410** |
| | 168 | 0.213 | 0.338 | **0.208** | **0.335** | 0.464 | **0.491** | **0.463** | **0.491** |
| | 336 | 0.235 | 0.360 | **0.208** | **0.357** | 0.499 | 0.517 | **0.496** | **0.513** |
| | 720 | 0.251 | 0.375 | **0.231** | **0.357** | **0.535** | **0.544** | 0.538 | **0.544** |
| | Avg. | 0.189 | 0.310 | **0.181** | **0.307** | 0.432 | 0.465 | **0.430** | **0.463** |

Table H.3: Results of univariate and multivariate time series forecasting for CoST.

## I INSTANCE-WISE VISUALIZATIONS

**Hard CL vs. Soft CL.** To assess the quality of instance-wise relationships captured by SoftCLT, we apply t-SNE (Van der Maaten & Hinton, 2008) to visualize the instance-wise representations, which are representations of whole time series obtained by max-pooling the representations of all time stamps, to both hard and soft CL. For this experiment, we apply our method to TS2Vec (Yue et al., 2022) with the UWaveGestureLibraryZ dataset from UCR archive (Dau et al., 2019). The results shown in Figure I.1 demonstrate that soft CL finds more fine-grained neighborhood relationships and distinguishes them better than hard CL.

**Embedding space vs. Input space.** To assess the relationship between the shape of time series and their positions in the embedding dimension, we employ t-SNE (Van der Maaten & Hinton, 2008) to embed instance-wise representations of time series using the InsectEPGRegularTrain dataset from UCR archive (Dau et al., 2019). Figure I.2 illustrates the results, with the left panel displaying the points in the embedding space and the right panel presenting line plots of the original TS. The colors of the points and lines are assigned based on the distances with their neighbors in the embedding space. From this figure, we observe that TS with the same color not only exhibit similar shapes, but also as the points in the embedding space move towards the upper right, the line plots of the original TS shift towards the upper left. This demonstrates that our method effectively captures detailed neighborhood relationships while maintaining alignment between the distances in the embedding space and the original input space.

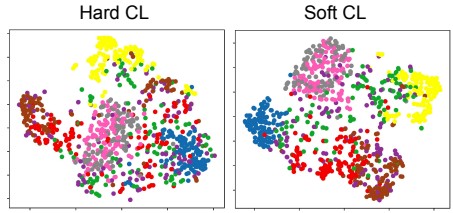

Figure I.1: Hard CL vs. Soft CL

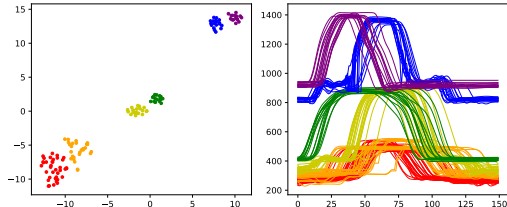

Figure I.2: Instance-wise visualizations

## J  TRANSFER LEARNING UNDER SEMI-SUPERVISED SETTINGS

In this study, we perform transfer learning in the semi-supervised settings using SleepEEG (Kemp et al., 2000) and EMG (Goldberger et al., 2000) datasets as the source and target datasets, respectively. Specifically, we apply our SoftCLT to TS-TCC under semi-supervised settings where we perform fine-tuning using partially labeled datasets. Figure J.1 presents the results, which indicate that by using only 10% of labeled data with the soft CL framework (red line), we are able to achieve an accuracy of 92.69%, which is approximately 15% higher than the accuracy obtained from the hard CL framework (blue line) under fully supervised settings. Furthermore, using only 50% of the labeled dataset allowed us to achieve 100% accuracy, whereas the state-of-the-art performance using fully labeled datasets is 97.56%.

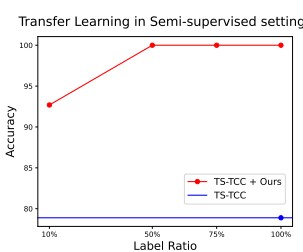

Figure J.1: TL results

## K  EFFECT OF DISTANCE METRICS BY TIME SERIES WITH VARYING LENGTH

We compare the average accuracy of 128 UCR datasets, where 11 datasets have varying time-length, and the other 117 datasets have the same time-length. As shown in Table K.1, DTW and TAM, both capable of comparing time series of different lengths using time warping, demonstrate better performance.

| Temporal CL | UCR datasets (Avg. Acc.(%)) | | | | | |
| --- | --- | --- | --- | --- | --- | --- |
| | Hard | | | Soft | | |
| TS length | Non-Varying (117/128) | Varying (11/128) | Total (128/128) | Non-Varying (117/128) | Varying (11/128) | Total (128/128) |
| COS | 84.8 | 72.6 | 83.7 | 85.7 | 75.0 | 84.7 |
| EUC | 85.1 | 73.3 | 83.9 | 85.8 | 73.9 | 84.8 |
| DTW | 84.8 | 73.6 | 83.9 | 85.9 | 75.2 | 85.0 |
| TAM | 85.0 | 73.4 | 83.9 | 85.9 | 75.3 | 85.0 |

Table K.1: Effect of DTW on time series with varying/non-varying length.

## L  DESIGN CHOICES FOR SOFT TEMPORAL CONTRASTIVE LEARNING

Various design choices can be considered for assigning soft labels in soft temporal contrastive learning. In this paper, we explore four different choices for the experiment, all of which assign high values to adjacent timestamps. Figure L.1 illustrates these four different choices. For **Neighbor**, **Gaussian**, and **Sigmoid**, we conduct a search for the optimal hyperparameter within the following range:

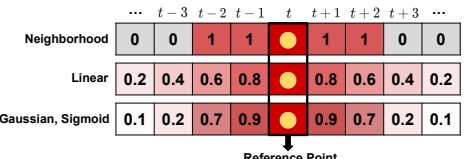

Figure L.1: Design for soft temporal CL

- **Neighbor**: A certain range within the reference point, with 10%, 30%, 50% of the sequence length.
- **Gaussian**: Standard deviation values of [0.5, 1.0, 1.5, 2.0, 2.5].
- **Sigmoid**: $\tau_T$ of [0.5, 1.0, 1.5, 2.0, 2.5].

## M    SOFT CONTRASTIVE LEARNING WITH NON-STATIONARY TIME SERIES

As our proposed soft instance-wise CL method generates labels by considering the distances between the *original* TS, global information from the entire TS is encapsulated within the representation of a single time step of that TS, which might enables to take account of non-stationarity, such as seasonality or distribution shifts present in the TS.

**Time series with seasonality.**    Figure M.1 displays a single TS of Adiac data from the UCR archive (Dau et al., 2019) and its visualization of temporal representations obtained from TS2Vec (Hard CL) and TS2Vec applied with our method (Soft CL). Note that obvious seasonal pattern are observed from the left panel of the figure. Each point in the right panel represents a representation of a single timestamp. The figure indicates that while hard CL fails to capture the seasonal patterns in the TS (as similar values, regardless of which phase they are in, are located closely), our proposed soft CL can grasp the global pattern, enabling it to capture the seasonal patterns (as similar values but with different seasonal phases are located differently).

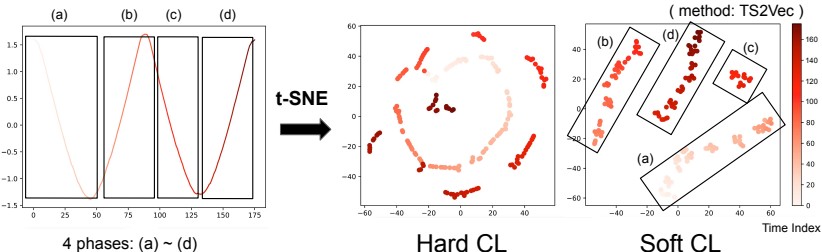

Figure M.1: Temporal visualization of TS with seasonality.

**Time series with distribution shift.**    Figure M.2 displays a single TS of EMD data from the UCR archive (Dau et al., 2019) and its visualization of temporal representations obtained from TS2Vec (Hard CL) and TS2Vec applied with our method (Soft CL). Note that distribution shifts exist in this data, and six different phases are observed from the left panel of the figure. Each point in the right panel represents a representation of a single timestamp. The figure indicates that while hard CL fails to capture the sudden change in the TS (as all points are located gradually regardless of the sudden change in the TS), our proposed soft CL can detect such distribution shifts (as points before/after a certain change are clustered in groups).

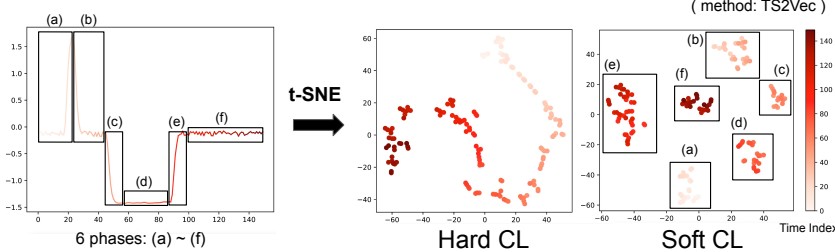

Figure M.2: Temporal visualization of TS with distribution shift.

## N   APPLYING SOFT CONTRASTIVE LEARNING TO TNC

In this section, we apply our SoftCLT to temporal neighborhood coding (TNC) (Tonekaboni et al., 2021), which employs temporal CL by leveraging the local smoothness of a signal's generative process to define neighborhoods, or positive pairs, in time with stationary properties. We apply our method to TNC by assigning soft temporal assignments based on the difference between the centroids of two time windows. Note that TNC does not include instance-wise CL, so we only apply soft temporal CL. To evaluate the effectiveness of this application, we conduct experiments using two datasets: the simulation dataset constructed in TNC and HAR [3]. The results are shown in Table N.1, demonstrating that SoftCLT applied to TNC improves performance on both datasets in terms of accuracy (Acc.) and AUPRC.

| | Simulation | | HAR | |
|---|---|---|---|---|
| | Acc.(%) | AUPRC | Acc.(%) | AUPRC |
| T-Loss | 76.66 | 0.78 | 63.60 | 0.71 |
| CPC | 70.26 | 0.69 | 86.43 | 0.93 |
| TNC | 97.52 | 0.996 | 88.21 | 0.940 |
| TNC + Ours | **97.95** | **0.998** | **89.14** | **0.952** |

Table N.1: Effect of soft temporal CL on TNC.

---

[3] https://archive.ics.uci.edu/dataset/240/human+activity+recognition+using+smartphones

