# OpenReview forum: "Soft Contrastive Learning for Time Series"
_ICLR.cc/2024/Conference — ICLR 2024 spotlight_

### Official Review · Reviewer_e9k3 · 2023-10-27

**Soundness:** 3 good
**Presentation:** 4 excellent
**Contribution:** 3 good
**Rating:** 8
**Confidence:** 4

**Summary:**

The authors propose a new contrastive learning method for time series. Specifically, they propose to remove the hard positive/negative assignment from the original NCE by a soft reweighting incorporating prior information about the temporal closeness or similarity of inputs. The authors evaluate their method on various time series-related tasks showing strong improvement compared to "hard" CL methods. They also provide ablation experiments concerning their new objective hyperparameters.

**Strengths:**

This paper is of high quality with the following strengths:

- Overall very well-written paper with an easy-to-follow structure, in particular:
    - Related work is structured and extensive (with the exception mentioned below in weaknesses).
    - Method is clear and figures relevant making the method easy to understand.
    - Experiments are very extensive and well described both in the manuscript and the supplementary materials.

- I appreciate that despite introducing numerous components and hyperparameters ( temperature, distance metric, weight function, etc..), authors provide ablations to each of these components.
- The authors discussed the additional computational complexity of their method and in particular DTW known to have a squared complexity.
- The amount of experiments carried out is very large and diverse

**Weaknesses:**

There are, however, some weaknesses, in particular in terms of related work, which I detail below:


**Related work**
As mentioned the related work is extensive but misses some seminal works regarding contrastive learning methods for time series tackling the challenges of inter/intra samples dependencies:
- First, "Subject-aware contrastive learning for biosignals" by Cheng et al (2020) proposes to only use negative representations from the same time series to "promote subject-invariant" representation, what the authors would refer to as "temporal CL".
- Second, "CLOCS: Contrastive Learning of Cardiac Signals Across Space, Time, and Patients" by Kiyasseh et al. (2021) proposes to on the contrary use representations from the same time series as positives, what the authors would refer to as "instance-wise CL". This is similar to TNC but with a neighborhood being defined as being from the same time series.
- Finally, and more importantly, "Neighborhood Contrastive Learning Applied to Online Patient Monitoring" by Yeche et al. (2021), introduces the trade-off used by the authors between instance-wise and temporal-wise CL controlled by $\lambda$. In particular, the objective proposed by Yeche et al., namely NCL, is similar to taking a hard assignment for instance-wise CL $w_I(i,j) = \mathbb{1}_{[i = j]}$ and a (discontinuous) uniform one over a window for temporal CL.

Thus, I think it's really important that this work refers to these three works and in particular NCL from which the SoftCLT is an extension to continuous neighborhood definitions. It would be nice to have a comparison to it as well.


**Clarity**

- The authors refer multiple times to instance-wise and temporal CL before defining it properly in the method section. I think pointing the reader to this section or defining the terms in the introduction could improve clarity.

- referring to a "temperature" parameter $\tau_t$ and $\tau_i$ can be quite misleading in the context of contrastive learning, where this term was coined by Chen et al. (2020), in the simCLR paper. (See my comment below on the choice of assignment function) Given the assignment function is some form of Laplacian kernel, referring to $l =\frac{1}{\tau}$ as a lengthscale parameter would be more coherent with literature and avoid confusion with temperature parameters from previous works on CL.

**Method**
- The authors define their assignment function around a sigmoid function which is defined over $\mathbb{R}$ whereas its input $D$ lies in $\mathbb{R}^+$. It seems to overcome this, they tweak around their sigmoid function to obtain a symmetric function $w(D) = \frac{2}{1+e^{Dt}}$. Why not rely on existing literature instead and typically use a Laplacian kernel $w(D) = e^{-\frac{D}{l}}$?
- Exploring further different kernel and their impact on performance would have been a nice addition. In particular, using a generalized Gaussian kernel and looking at the impact of the shape parameter $\beta$ would be nice as $\beta=1$ is SoftCLT and  $\beta=\infty$ is NCL temporal CL.
- Exploring further the impact of the trade-off between local (temporal) and global (instance) features learning ruled by $\alpha$ would be a nice addition to ablations.
**Conclusion**


Clarity and Method weakness are easily addressable. Regarding related work, despite the similarities with NCL, I still think the contribution to be significant given the novelty around the neighborhood/assignment function and the extent of the experiments on various tasks, justifying my choice of recommending acceptance. However,  I firmly believe the three works I mentioned should be correctly cited in particular the link to Yeche et al. (2021) work.

**Questions:**

I don't have any questions beyond the points raised in the above sections.

---

> ### Author Response · Authors · 2023-11-19
>
> **W1. Related works from medical domain**
>
> In our initial submission, we cited papers about general TS rather than specific domains like the medical domain. However, we agree that a discussion on related works from specific domains would still be interesting and helpful to audiences. We have added a brief discussion on the papers that e9k3 suggested in **Section 2**. Below is a longer discussion that e9k3 might be specifically interested in:
>
> - **Subject-aware contrastive learning for biosignals (Cheng et al, 2020)**:  Their proposed subject-aware CL contrasts multiple TS, which can be considered as **instance-wise CL** rather than temporal CL, as instance-wise CL contrasts the representations of **multiple TS**, while temporal CL contrasts the representations of multiple timestamps within a **single TS**, as illustrated in **Figure 1**. By architecture design, subject-aware CL cannot perform temporal CL, as they generate representations for the entire TS, such that the information along with timestamps are entangled.
>
> - **CLOCS: Contrastive Learning of Cardiac Signals Across Space, Time, and Patients (Kiyasseh et al, 2021)**: Their proposed CLOCS performs CL over temporal and spatial dimensions, where the concept of spatial dimension is close to the channel (or multivariate) dimension in general TS. Hence, this idea might have a limited applicability, e.g., for univariate TS.
>
> - **Neighborhood Contrastive Learning Applied to Online Patient Montoring (Yeche et al, 2021)**:  NCL performs CL with **multiple hard** assignments, where assignments are based on predefined attributes of instances. This assignment strategy of NCL is similar to NNCLR [A]. NCL jointly optimizes two conflicting losses with a trade-off: the neighbor alignment loss maximizing the similarity of neighbors as well as positive pairs, and the neighbor discriminative loss maximizing the similarity of positive pairs while minimizing the similarity of neighbors. However, different from two conflicting losses in NCL, our proposed two losses (soft instance-wise CL loss and soft temporal CL loss) do not contradict each other and operate on different dimensions: instance-wise and temporal, respectively.
>
> Also, similar to subject-aware CL, **NCL does not perform temporal CL**; it contrasts representations of multiple TS, which corresponds to instance-wise CL. Note that, as illustrated in Figure 5 of the NCL paper, representations obtained from NCL disregard the temporal axis. Hence, by architecture design, subject-aware CL cannot perform temporal CL, as they generate representations for the entire TS, such that the information along with timestamps are entangled.
>
> As we understand, NCL can be considered a variation of NNCLR [A], where it takes advantage of supervision from the medical domain to determine neighbors and jointly optimizes two conflicting losses. Hence, we categorize NCL as a soft contrastive learning and have added a discussion in **Section 2**.
>
> [A] Dwibedi, Debidatta, et al. "With a little help from my friends: Nearest-neighbor contrastive learning of visual representations." ICCV (2021)
>
>
> &nbsp;
>
> **W2-a. Definition of instance-wise and temporal CL**
>
> Thanks for the suggestion, we have added the definition of instance-wise and temporal CL in **Section 1** and **Figure 1**. We believe this will improve the clarity of the proposed method from the earlier part of our paper.
>
>
> &nbsp;
>
> **W2-b. Confusion with the term "temperature"**
>
> Thanks for raising this concern. We agree that it can be confused with the temperature parameter used in CL literature. To avoid any confusion and maintain coherence with previous works, we replaced the term "temperature" with "hyperparameter controlling the sharpness" for $\tau$ in our paper.

---

> > ### Author Response · Authors · 2023-11-19
> >
> > **W3-a. Laplacian kernel instead of sigmoid for instance-wise CL**
> >
> > & **W3-b. Exploring different kernels for assignment function**
> >
> > Indeed, we also conducted an ablation study on the choice of the assignment function for instance-wise CL in **Table F.1** (to make sure, the one in **Table 6(b)** is for temporal CL). However, as the Laplacian kernel was initially not our candidate, we additionally conducted experiments using the **Laplacian kernel** for instance-wise CL. As shown in the updated **Table F.1**, the performance of the Laplacian kernel (83.1%) is lower than the best performance of the sigmoid function (83.9%).
> >
> > Though we have not tried generalized Gaussian variants with different $\beta$, we compared other types of choices like Neighbor or Linear for temporal CL (where the details can be found in **Appendix L**) instead, as different types of assignment functions would show more different results.
> >
> > We note that the details of assignment functions can be found in below:
> > - Temporal CL: **Table 6(b)**, **Appendix  L**
> > - Instance-wise CL: **Appendix  F**
> >
> > By the way, we note that the role of the assignment function is not mainly on squashing the range of D to [0,1] for instance-wise CL, as D is the min-max normalized distance in this case; we have updated the definition of D in **Section 3.2** accordingly. However, the assignment function is still useful in that it highlights a certain range of distances by nonlinear transformation, depending on the choice of the assignment function. To confirm this, we also added the case when we omit the assignment function (w/o kernel) in **Table F.1**, which shows worse performance (79.1%) than others.
> >
> > &nbsp;
> >
> >
> > **W3-c. Trade-off between temporal and instance-wise contrastive loss**
> >
> > The hyperparameter controlling the trade-off between temporal and instance-wise contrastive losses is chosen by validation; in fact, $\lambda=0.5$ is mostly the best, and this observation is consistent with the original paper of methods that we apply SoftCLT on top of.
> >
> > We note that, different from two conflicting losses in NCL, our proposed two losses (soft instance-wise CL loss and soft temporal CL loss) do not contradict each other and operate on different dimensions: instance-wise and temporal, respectively. As shown in the updated **Figure 1**, they construct negative pairs differently; temporal CL constructs negative pairs from the same TS but of different timestamps, while instance-wise CL constructs negative pairs from different TS but of the same timestamp.

---

### Official Review · Reviewer_mvHV · 2023-10-27

**Soundness:** 2 fair
**Presentation:** 3 good
**Contribution:** 2 fair
**Rating:** 6
**Confidence:** 4

**Summary:**

This study introduce a new method of performing constrastive learning, a soften version of normal positive-negative strategy. These soft assignments are determined by the distance between time series in the data space for instance-wise contrastive loss and the difference in timestamps for temporal contrastive loss.

**Strengths:**

Contribution:

- The idea of soft constrastive learning is straight-forward and natural. The underlying functions are widely-adopted and straightforward to implement.
- The experiments are extensive and cover many time-series tasks (classification, anomaly detection) as well as scenario (self/semi supervised and supervised learning). The comparison with soft-CL techniques from other domains and ablation study make the whole experimental section be quite well-rounded.

Representation:

- Intuitive and direct illustrations via Figures (e.g. Fig.1, 2)

**Weaknesses:**

- Contribution:
    - For instance-wise CL:
        - the use of DTW might be a potential bottleneck in case of dealing with lengthy time-series. While the authors suggest the use of FastDTW, the complexity regarding the memory might be increased, and also the potential reduce in approximation (in case the warping path between two time series instances is highly nonlinear). In other words, the choices of DTW or FastDTW are hurting the pipeline in some ways.
        - the calculation of weight based on the distance in the data space. However, this make the weighting process be dependent on the scale of input data. Together with the wrapper of Sigmoid function, it might be saturated upon too large or too small input. This effect might make the weights not representative to use in instance-wise CL. While empirically, it illustrates the effective over in latent space, more effort need to be done to consider on which space one should rely on to calculate distance.
    - For temporal-wise CL, the current weight assignment implicitly assume the data from neighbors’ timesteps should be weighted heavier than the data from far timesteps. However, that behavior might not always hold true, as illustrated in work of Tonekaboni (2021).

**Questions:**

The authors please address or provide answers to any questions from the weaknesses listed.

---

> ### Author Response · Authors · 2023-11-19
>
> **W1-a. Complexity issue with DTW and FastDTW**
>
> We first emphasize that DTW is just one option, and other distance metrics can be employed; we chose DTW because it performed best according to **Table 6(d)**. We note that other simpler metrics without dynamic programming like the cosine distance (COS) or Euclidean distance (EUC) also show decent performance, so they can be the option if the complexity matters.
>
> Nonetheless, regarding the concern on the computational complexity, we note that distances between TS (for soft assignments for the instance-wise CL) are computed based on the original TS, such that they can be precomputed offline before training or cached for efficiency.
>
> Regarding FastDTW, we found that the approximation error is almost negligible for datasets we experimented with, resulting in identical performance throughout experiments.
>
> For better clarification, we added more description in **Section 3.2**.
>
> &nbsp;
>
> **W1-b. Soft assignment's dependency on the scale of the input data**
>
> Thank you for bringing this to our attention. In fact, we missed to talk about the detail that we **min-max normalized the pairwise distance matrix**, such that the minimum and maximum values become zero and one, respectively. We have updated the definition of D in **Section 3.2** accordingly.
>
> We also note that the specific details that we could miss should be clear through the code we provided in the supplementary materials, and we will release the code upon acceptance for reproducibility.
>
> &nbsp;
>
>
>
> **W2. Implicit assumption on soft assignments for temporal CL**
>
> (Tonekaboni, 2021) considers CL for TS under non-stationarity. Indeed, non-stationarity can be addressed by other strategies like TNC [A] or CoST [B], and our method can be applied on top of them to improve the performance further. For example, CoST [B] is designed to learn disentangled seasonal-trend representations, and applying our SoftCLT on top of CoST improves the performance, which can be found in the new experiment in **Table H.3 in Appendix H**.
>
> [A] Tonekaboni, Sana, Danny Eytan, and Anna Goldenberg. "Unsupervised representation learning for time series with temporal neighborhood coding." ICLR (2021)
>
> [B] Woo, Gerald, et al. "CoST: Contrastive learning of disentangled seasonal-trend representations for time series forecasting." ICLR (2022)
>
>
> Nonetheless, we provide more discussion on our method with non-stationarity below.
>
>
>
> &nbsp;
>
>
> **W2-a. TS with non-stationarity - (1) Seasonality**
>
> Soft temporal CL assigns more weight to closer timestamps, so one might be concerned if it fails when seasonality is present.
> We argue that **instance-wise CL can indirectly capture the seasonality of TS**, as it contrasts by looking at the representations of the original TS, while **temporal CL** takes advantage of the non-seasonal portions. (In **Section 3.2**, we have clarified that the soft assignments for instance-wise CL are calculated using the original TS, not the augmented views.)
> Following the terms from the review of e9K3, soft temporal CL and soft instance-wise CL captures the local and global features, respectively.
>
> &nbsp;
>
> To support our claim, we present both qualitative and quantitative analyses as follows.
>
>
> **Quantitative analysis (Table 8)**
>
> **Table 8** categorizes 128 UCR datasets by seasonality and shows that the performance gain by sot temporal CL is consistent regardless of the degree of seasonality.
>
> &nbsp;
>
> **Qualitative analysis (Section M - Figure M.1)**
>
> We conducted additional analysis to show that our proposed **soft CL captures seasonality better than the hard CL**, as shown in **Figure M.1**. This figure depicts the t-SNE of representations for all time steps in a single TS **with seasonality**. The figure indicates that while hard CL fails to capture the seasonal patterns in the TS (as similar values, regardless of which phase they are in, are located closely in the embedding space), our proposed soft CL can grasp the global pattern, enabling it to capture the seasonal patterns (as similar values but with different seasonal phases are located differently in the embedding space).
>
>
> &nbsp;
>
>
> **W2-b. TS with non-stationarity: distribution shift**
>
> Soft instance-wise CL captures the global view of the TS, and we believe this allows us to detect the **distribution shifts** in the TS. To support this claim, we present additional analysis in **Figure M.2 in Appendix M**.
>
> **Figure M.2** shows the t-SNE of representations for all timestamps in a single TS **with distribution shifts**. The figure indicates that while hard CL fails to capture the sudden change in the TS (as all points are located gradually regardless of the sudden change in the TS), our proposed soft CL can detect such distribution shifts (as points before/after a certain change are clustered in groups).

---

> > ### Comment · Reviewer_mvHV · 2023-11-21
> > **Additional Review of Reviewer mvHV**
> >
> > Dear Authors,
> >
> > Thanks for making clarifications and great attempt in modifying the manuscript.
> > However,  I still have several concerns basing on top of your answers:
> > - *we found that the approximation error is almost negligible for datasets we experimented with*
> >
> > I would expect a verification for this claim, I suggest the Authors providing an additional experiment to your Ablation study in Table 6(d).
> >
> > - *our method can be applied on top of them to improve the performance further*
> >
> > I suspect this claim in case of incorporating softCL with TNC [1].
> > To clear out my initial comment, I think the current weighting strategy of temporal alignment contrasts with what proposed in [1]. I am not sure if the two method can yield better result if being fused together.
> >
> > In addition, why it is more ideal if your framework is more sensitive toward seasonality? For example in Figure M.1, why phase (a) and (c) representations need to be distinguished, as illustrated by the figure, what if that two phase have the same label?
> >
> > To sum up, with these concerns, for now, I still keep the scores as it is.
> >
> >
> >
> >
> > [1] Tonekaboni, Sana, Danny Eytan, and Anna Goldenberg. "Unsupervised representation learning for time series with temporal neighborhood coding." ICLR (2021)

---

> ### Author Response · Authors · 2023-11-21
>
> Thank you for engaging more in the discussion!
>
> Below we provide our responses to each concern, and we updated our manuscript accordingly (Including the new **Appendix N** and minor changes):
>
> &nbsp;
>
> > (1) Your request to verify our statement that the FastDTW approximation error is negligible
>
> First of all, we would like to clarify that concerns you might have around DTW and FastDTW like *how much DTW is computationally expensive* and *how accurate the approximation by FastDTW is* are **out of scope** for this work, as **our main contribution does not lie in the particular design choices on a specific metric (e.g., DTW), but in the proposal of a simple yet effective soft contrastive learning framework that harnesses (self-)supervision from the data space.** If you really concern on the complexity of DTW or inaccuracy of FastDTW, then you can go for the simple Euclidean or cosine distance at the cost of small performance drops; according to **Table 2 and 6(d)**, the performance drop is indeed minor compared to the gain from the baseline: in classification, **TS2Vec: 83.0% vs. Ours-EUC: 84.8% vs. Ours-DTW: 85.0%**.
>
> &nbsp;
>
> Nonetheless, we address this concern as follows. As we did not keep our early experiments with FastDTW, we are running more experiments now, and below we provide a part of results from the ongoing experiments with 60 out of 128 UCR datasets. Note that we used python libraries to compute distances: `tslearn` for DTW and `fastdtw` for FastDTW.
>
> &nbsp;
>
> **[Concern 1] Difference in soft assignments ($w_I$).**
> The average L2 error between the soft assignments generated by DTW and FastDTW over 60 datasets is 0.0019. Note that the soft assignments lie in the range [0,1], so **0.0019 is relatively a small error**.
>
> &nbsp;
>
> **[Concern 2] Difference in accuracy.**
> The average accuracy difference in SoftCLT when using DTW and FastDTW is **0.04%p, which is negligible** for our comparisons using only one decimal place. Notably, among the 60 datasets, 53 have exhibited exactly the same accuracy. Based on these results, we can say that using FastDTW as an approximation of the original DTW is acceptable.
>
>
> &nbsp;
>
> > (2) TNC + SoftCLT; Doesn't TNC contradict with the soft temporal CL?
>
> Thanks for your suggestion, we have added the TNC + SoftCLT experiment in the new **Appendix N**.
>
> The proposed soft assignment strategy can be applied to TNC. First note that TNC finds temporal neighbors and performs contrastive learning along with the temporal dimension. **Instead of hard assignments, SoftCLT applies soft assignments to those neighbors found by TNC based on the temporal distance.** Note that TNC performs only temporal contrasting, so we soften the temporal dimension only.
>
>
> &nbsp;
>
> To observe the effectiveness of SoftCLT on top of TNC, we implemented our SoftCLT on the official TNC code and experimented on the Simulation and HAR datasets (Table 2 in TNC; the ECG dataset is too large to quickly run the experiments). As in **Table N.1** in the new revision or the table below, **applying SoftCLT to TNC results in consistent performance improvements across datasets.**
>
>
>
> | | Simulation | | HAR | |
> |-|-|-|-|-|
> | | Acc.(%) | AUPRC | Acc.(%) | AUPRC |
> |T-Loss | 76.66 | 0.78 | 63.60 | 0.71|
> |CPC| 70.26 | 0.69  | 86.43 | 0.93|
> |TNC| 97.52 | 0.996  | 88.21 | 0.940 |
> |TNC+Ours| 97.95 | 0.998 | 89.14 | 0.952 |
>
> &nbsp;
>
>
> Finally, note that TNC is a relatively early work compared to other methods, such that its performance is generally lower than others (e.g., in classification, TNC: 76.1% vs. TS2Vec: 83.0%), making it low-prioritized in the application of our method throughout experiments in our main paper.
>
> &nbsp;

---

> > ### Author Response · Authors · 2023-11-21
> >
> > > (3) Why it is more ideal if your framework is more sensitive toward seasonality? For example in **Figure M.1**, why phase (a) and (c) representations need to be distinguished, as illustrated by the figure, what if that two phase have the same label?
> >
> >
> > First of all, we would like to clarify that **we do not claim that being more sensitive toward seasonality is more ideal**. Also, the purpose of Appendix M is to share our empirical observations on how embedding is different when learned with hard CL or soft CL, that embedding vectors learned with soft CL are 1) closer if they are on a similar phase in seasonality, and 2) clustered in different places if distribution shifts exist. We are happy to discuss anything presented in our paper, but we believe a concern on the experimental results we provided for your information in the appendix should not be the reason for rejecting a paper.
> >
> > &nbsp;
> >
> > Regarding "why phase (a) and (c) representations need to be distinguished", being in the same phase of seasonality does not imply that being embedded in the same space is ideal. There might be several reasons why they are distinguished (though we do not claim that they have to do so). For example, **the given time series does not exhibit perfect seasonality** (by looking at the relative position of each peak and the horizontal lines of black bounding boxes), i.e., the characteristics other than seasonality (e.g., trend or noise) can cause to distinguish (a) and (c) in **Figure M.1**. However, as the seasonality in the given time series looks strong, so it is reasonable to place them close to each other, as illustrated in **Figure M.1**; i.e., (a) and (c) are close to each other and (b) and (d) are close to each other.
> >
> > &nbsp;
> >
> > We hope now all your concerns are successfully addressed and/or you understand they are beyond the scope of this work. If you still have some concerns and/or suggestions, please share with us. We will try to address them before the end of the discussion phase.

---

> > > ### Comment · Reviewer_mvHV · 2023-11-21
> > > **Final Response From Reviewer mvHV**
> > >
> > > Dear Authors,
> > >
> > > Thank you for the further clarifications.
> > >
> > > Your answer this time seems fair enough for me.
> > >
> > > I have modified my final score for this work.
> > >
> > > Thanks.
> > >
> > > Best,

---

### Official Review · Reviewer_EHZC · 2023-10-30

**Soundness:** 3 good
**Presentation:** 3 good
**Contribution:** 2 fair
**Rating:** 6
**Confidence:** 4

**Summary:**

This study argues that when using time series data in contrastive learning, contrasting ( between positive and negative) instances or values located in proximity can lead to the neglect of their inherent correlation. Therefore, they introduce a continuous (referred as soft) weighting approach as an alternative to binary labeling, serving as a generalization of the standard contrastive loss, with the transformation occurring when replacing soft assignments with hard assignments of zero for negatives and one for positives. For soft assignment, the authors take into account two aspects: the similarity between two time series in data space and the proximity of two time series with respect to their timestamps.

**Strengths:**

The papers is well written and clear. The figures presented help to clarify the main idea and how it is implemented. The idea is novel for the simplified setup that is considered. The experimental results cover 3 downstream tasks and comprehensively evaluate the assumed setup.

**Weaknesses:**

The paper addresses a simplified scenario in which issues related to noise, seasonality, and non-stationarity are not considered, as there is no apparent mechanism in the approach to address these prevalent challenges found in real-world time series data.
Regarding robustness in the presence of noise and non-stationarity there  is no specific discussion or empirical evaluation.  Regarding seasonality, the authors mentioned "Our conjecture is that TS in the real world usually do not exhibit the perfect seasonality, as indicated by the ADF test result, such that SoftCLT takes advantage of the non-seasonal portions." While perfect seasonality may be absent in some datasets and may vary in intensity across different datasets, I believe completely disregarding it is not a practical approach.

**Questions:**

Can this case be elaborated a bit further:” when α = 1, we give the assignment of one to the pairs with the distance of zero as well as the pairs of the same TS” What if in the same TS we are experiencing two different patterns, shifts or different distribution

In equation 3, augmentation for I and i+N, how it  is performed? What if there is only a shift in the pattern in the instances, otherwise there are very similar how you address this in your computation, It would be great to include an illustration for this case to show you approach is robust to shift (or some noise) which is very common in real world applications.


How do you manage non stationary in the time series, where the immediate next point might be the start of a different distribution? How your similarity comparison handles it when the the proximity is assumed to have high similarity which is not necessarily true.

---

> ### Author Response · Authors · 2023-11-19
>
> **W1. TS with noise, seasonality, and non-stationarity**
>
> We found that concerns raised by EHZC are mostly about specific problems in TS, such as noise, seasonality, and non-stationarity. However, we note that **the primary focus on this work is on improving CL for TS** (which is SOTA in TS representation learning), **rather than addressing specific problems in TS**. Indeed, our proposed **SoftCLT is a simple** yet effective soft contrastive learning strategy for TS, so addressing such problems at the expense of simplicity might result in hampering our main contribution.
>
> Rather, such specific problems can be addressed by other strategies, and our method can be applied on top of them to improve the performance further. For example, CoST [A] is designed to learn disentangled seasonal-trend representations, and applying our SoftCLT on top of CoST improves the performance, which can be found in the new experiment in **Table H.3 in Appendix H**.
>
> Nonetheless, we discuss each problem below.
>
> [A] Woo, Gerald, et al. "CoST: Contrastive learning of disentangled seasonal-trend representations for time series forecasting." ICLR (2022)
>
> &nbsp;
>
>
>
> **W1-a. TS with noise**
>
> We agree that some significant noises could pose challenges when calculating distances between TS for soft instance-wise CL. However, we think addressing this issue falls beyond our current scope and is more aptly addressed by metrics designed for measuring distances between TS, as we are not confined to a specific metric. Data preprocessing techniques such as smoothing may alleviate this concern.
>
> &nbsp;
>
>
>
> **W1-b. TS with non-stationarity - (1) Seasonality**
>
> Soft temporal CL assigns more weight to closer timestamps, so one might be concerned if it fails when seasonality exists.
>
> We argue that **instance-wise CL can indirectly capture the seasonality of TS**, as it contrasts instances by looking at the representations of their original TS, while **temporal CL** takes advantage of the non-seasonal portions. (In **Section 3.2**, we have clarified that the soft assignments for instance-wise CL are calculated using the original TS, not the augmented views.)
> In other words, following the terms from the review of e9K3, soft temporal CL and soft instance-wise CL captures the local and global features, respectively.
>
> &nbsp;
>
> To support our claim, we present both qualitative and quantitative analyses as follows.
>
> **Quantitative analysis (Table 8)**
>
> **Table 8** categorizes 128 UCR datasets by seasonality and shows that the performance gain by soft temporal CL is consistent regardless of the degree of seasonality.
>
> &nbsp;
>
> **Qualitative analysis (Section M - Figure M.1)**
>
> We conducted additional analysis to show that our proposed **soft CL captures seasonality better than the hard CL**, as shown in **Figure M.1**. This figure depicts the t-SNE of representations for all time steps in a single TS **with seasonality**. The figure indicates that while hard CL fails to capture the seasonal patterns in the TS (as similar values, regardless of which phase they are in, are located closely in the embedding space), our proposed soft CL can grasp the global pattern, enabling it to capture the seasonal patterns (as similar values but with different seasonal phases are located differently in the embedding space).
>
>
> &nbsp;
>
>
> **W1-c, Q3. TS with non-stationarity: distribution shift**
>
> To our understanding, your concern around "non stationary in the time series, where the immediate next point might be the start of a different distribution" is the problem of distribution shift. Soft instance-wise CL captures the global view of the TS, and we believe this allows us to detect the **distribution shifts** in the TS. To support this claim, we present additional analysis in **Figure M.2 in Appendix M**.
>
> **Figure M.2** shows the t-SNE of representations for all timestamps in a single TS **with distribution shifts**. The figure indicates that while hard CL fails to capture the sudden change in the TS (as all points are located gradually regardless of the sudden change in the TS), our proposed soft CL can detect such distribution shifts (as points before/after a certain change are clustered in groups).

---

> > ### Author Response · Authors · 2023-11-19
> >
> > **Q1. What if a pair of views from the same TS instance have two different patterns (shifts or different distribution)?**
> >
> > We note that two views are generated by **data augmentation (DA)**, where DA is meant to generate multiple data with different raw values but same semantics in machine learning. Hence, DA should not destroy the semantics; in other words, **DA must not generate two different patterns for an instance if the patterns should be recognized as the characteristics of different classes**. In our case, we do not prescribe any specific DA techniques, but follow DA used in the baseline where our SoftCLT is applied.
> >
> > If we misunderstood your question and/or our answer is insufficient, please ask us again.
> >
> >
> > &nbsp;
> >
> >
> > **Q2. How is augmentation for i and i+N performed**
> >
> > Note that i and i+N are indices of two augmented views from the same instance. As described in our answer for Q1, our proposed SoftCLT is not constrained to any specific architecture or any form of DA, but follows the settings of the target CL methods that we apply SoftCLT. Regarding the concern on shift or some noise potentially introduced by DA, such DA should not be used if they destroy the semantics.
> >
> > As we applied our SoftCLT to TS2Vec [A], TS-TCC [B], and CA-TCC [C], we provide DA techniques used in these methods below:
> >
> > - TS2Vec: It generates two views of a single TS by randomly sampling two segments ***with an overlap***, and it contrasts only the overlapped parts.
> > - TS-TCC and CA-TCC: They generate two views of a single TS by two different augmentations: strong augmentation (permutation-and-jitter) for one and weak augmentation (jitter-and-scale) for the other.
> >
> > To clarify this, we have included additional details on the data augmentation techniques we employed in **Section 4**. Thank you for bringing this to our attention.
> >
> >
> > [A] Yue, Zhihan, et al. "Ts2vec: Towards universal representation of time series." AAAI (2022)
> >
> > [B] Eldele, Emadeldeen, et al. "Time-series representation learning via temporal and contextual contrasting." IJCAI (2021)
> >
> > [C] Eldele, Emadeldeen, et al. "Self-supervised contrastive representation learning for semi-supervised time-series classification." TPAMI (2023)

---

> > > ### Comment · Reviewer_EHZC · 2023-11-22
> > >
> > > Thanks for addressing my questions and performing further experimentation. In my opinion, the novelty presented in this work is marginal and I will keep the score I have given before.

---

### Official Review · Reviewer_bKG2 · 2023-10-31

**Soundness:** 2 fair
**Presentation:** 3 good
**Contribution:** 2 fair
**Rating:** 6
**Confidence:** 5

**Summary:**

The authors address challenges in time series data annotation and the limitations of standard contrastive learning (CL) in representing TS data. Key contributions of the paper are:

- Introduction of SoftCLT, a soft contrastive learning strategy tailored for time series data. Their framework can be adapted to other CL frameworks relatively easily.
- The proposal of soft contrastive losses for both instance and temporal dimensions, addressing the shortcomings of existing CL methods for TS.
- Comprehensive experimental evidence demonstrating that SoftCLT enhances state-of-the-art performance across multiple TS tasks.

**Strengths:**

The submission has the following strenghts:

- Ablation study is present and seems to demonstrate the usefulness of the proposed additions.
- Compared to the selected baselines (emphasis on selected), the model performs well.
- I appreciate that the authors have chosen to go for a more detailed analysis of the representation learning abilities of their model. By this I mean that rather than considering only task-performance, they also investigate aspects such as robustness to non-stationarity, and also semi-supervised learning. This is usually absent from related papers.
- The ideas are well explained, the paper is clear.

**Weaknesses:**

The submission has the following weaknesses:
- Problem with the comparisons. Entirely absent from the main paper is any comparison with CoST [1], or any more recent contrastive approach. While this is a single issue it is one I am quite concerned about. Similarly, a comparison to recent approaches in the regression setting of TS2Vec (nothing prevents that comparison, the TS2Vec code works seamlessly for both approaches). My reasoning is that the proposed idea is interesting, but also relatively simple. This is fine in general: simple ideas bring value in research as well. However, coupled with a lack of comparison to recent approaches, it is very difficult to ascertain the value of the contribution.

Currently this is enough for me to not recommend acceptance, but as noted in the questions section, I am willing to update my score should the authors adress this.

References
[1] CoST: Contrastive Learning of Disentangled Seasonal-Trend Representations for Time Series Forecasting

**Questions:**

As noted in the Weaknesses section, I would like a detailed comparison with CoST and an evaluation in the regression setting. If the authors provide this, and the results warrant it, I will raise my score.

---

> ### Author Response · Authors · 2023-11-19
>
> **W1, Q1. Detailed comparison with CoST and an evaluation in the regression setting**
>
> Thanks for bringing this to our attention. We applied our SoftCLT to CoST [A], and provide the results below and **Table H.3 in Appendix H** in the revision:
>
> | | Multivariate || forecasting | | Univariate | | forecasting | |
> | ---- | ---- | ---- | ---- | ---- | ---- | ---- | ---- | ---- |
> | | CoST | | CoST + Ours | | CoST | | CoST + Ours | |
> | | MSE | MAE | MSE | MAE | MSE | MAE | MSE | MAE |
> | ETTh1 | 0.650 | 0.585 | **0.619** | **0.570** | 0.094 | 0.230 | **0.091** | **0.227** |
> | ETTh2 | 1.322 | 0.876 | **1.315** | **0.872** | 0.161 | 0.307 | **0.156** | **0.304** |
> | ETTm1 | 0.409 | 0.439 | **0.407** | **0.436** | 0.054 | 0.164 | **0.051** | **0.159** |
> | Weather | 0.432 | 0.465 | **0.430** | **0.463** | 0.189 | 0.310 | **0.181** | **0.307** |
>
> We observed consistent improvements in performance for CoST in both multivariate and univariate forecasting tasks under various datasets. The results are average MSE/MAE above 5 horizons, and averaged over four runs with different random seeds, using the official code of CoST.
>
> [A] Woo, Gerald, et al. "CoST: Contrastive learning of disentangled seasonal-trend representations for time series forecasting." ICLR (2022)
>
> &nbsp;
>
> **W2. Comparison with recent contrastive approaches**
>
> We agree that comparing with recent CL approaches and figuring out the applicability of our SoftCLT to them should be interesting and strengthen our contribution. We are supposed to apply our SoftCLT to the SOTA methods per downstream tasks, e.g., TS2Vec [A] for classification/anomaly detection and TS-TCC/CA-TCC for semi-supervised learning. Thanks to bKG2, we have added CoST [B] for regression following their suggestion.
>
> To be more specific, we provide how we choose the baseline that we apply SoftCLT to:
>
> 1. **Classification (TS2Vec [A])**: We chose the widely used UCR 128 and UEA 30 datasets for both univariate and multivariate TS classification, respectively, as these datasets cover TS from various domains and lengths, and TS2Vec has demonstrated SOTA performance for these tasks.
>
> 2. **Semi-supervised Classification (TS-TCC [B], CA-TCC [C], TS2Vec [A])**: We chose CA-TCC, as it is the CL method specifically designed for semi-supervised settings and has demonstrated SOTA performance. Given that CA-TCC is an extension of TS-TCC, we also included TS-TCC in our evaluation. Additionally, TS2Vec, renowned for its SOTA performance in standard classification, is also included.
>
> 3. **Transfer Learning (TS-TCC [B], CA-TCC [C])**: While there were other candidates (TS2Vec [A], TF-C [D], CoST [E], CLOCS [F]), we specifically chose TS-TCC and CA-TCC for the following reasons:
>
> - TS2Vec [A]: Its performance, as indicated by experiments conducted by CA-TCC, is mostly worse than both TS-TCC and CA-TCC.
>
> - TF-C [D]: Their result is not reproducible with their official code. This issue seems not unique to our experience by looking at several GitHub issues.
>
> - Other CL methods (CoST [E], CLOCS [F]): These methods are specifically designed for TS forecasting, hence not considered for transfer learning in classification. Their performance is inferior to TS-TCC, as shown in the updated **Table 4(a)**.
>
> 4. **Anomaly Detection (TS2Vec [A])**: Given the scarcity of CL methods addressing anomaly detection, TS2Vec, a SOTA method in this domain, is chosen for our evaluation.
>
>
> We have also added other recent methods (CoST [E], CLOCS [F], LaST [G]) to **Table 4** by comparing the transfer learning performance, as it would be beneficial to audiences.
>
>
> [A] Yue, Zhihan, et al. "Ts2vec: Towards universal representation of time series." AAAI (2022)
>
> [B] Eldele, Emadeldeen, et al. "Time-series representation learning via temporal and contextual contrasting." IJCAI (2021)
>
> [C] Eldele, Emadeldeen, et al. "Self-supervised contrastive representation learning for semi-supervised time-series classification." TPAMI (2023)
>
> [D] Zhang, Xiang, et al. "Self-supervised contrastive pre-training for time series via time-frequency consistency." NeurIPS (2022).
>
> [E] Woo, Gerald, et al. "CoST: Contrastive learning of disentangled seasonal-trend representations for time series forecasting." ICLR (2022)
>
> [F] Kiyasseh, Dani, Tingting Zhu, and David A. Clifton. "Clocs: Contrastive learning of cardiac signals across space, time, and patients." ICML (2021).
>
> [G] Wang, Zhiyuan, et al. "Learning latent seasonal-trend representations for time series forecasting." NeurIPS (2022)

---

> > ### Comment · Reviewer_bKG2 · 2023-11-22
> > **Response to the rebuttal**
> >
> > I thank the reviewers for adressing my point about CoST. As this adresses the main conern I have, I have raised my score to a 6.

---

### Author Response · Authors · 2023-11-19

In what follows, we summarize additional experiments we provided during the rebuttal period:

(**Table F.1**; e9k3) Laplacian kernel for instance-wise CL

(**Table H.3**; bKG2) Applying SoftCLT to CoST for TS forecasting

(**Figure M.1,M.2**; EHZC, mvHV) Soft contrastive learning with non-stationary TS data

&nbsp;

For your convenience, we uploaded the pdf file containing both the revised manuscript and appendices. Also, in our revised manuscript, we **highlighted the major changes in blue.**

We believe all the discussions and empirical results are valuable and would further strengthen our contribution. **If we missed something or if you have further questions or suggestions, please share them with us**; we are happy to address them.

&nbsp;

Thank you very much.

Authors.

---

### Author Response · Authors · 2023-11-19

# General Comment

We sincerely thank you for dedicating your time and effort to reviewing our paper.

Our work proposes SoftCLT(Soft Contrastive Learning for Time Series), a soft contrastive learning framework tailored for time series data that operates in both temporal dimension and instance-wise dimension and can be easily adapted to other CL methods.
As highlighted by the reviewers, our work is recognized as being well-written and easy to follow (bKG2, EHZC, e9k3), novel (EHZC), straightforward (bKG2, mvHV, e9k3), and adaptable to other methods (bKG2, mvHV). It demonstrates SOTA performances across four different TS tasks (all) with detailed analyses (bKG2, e9k3).

&nbsp;

In our responses, we've attended to the concerns raised by all reviewers, augmenting our claims with additional experiments and analyses. Below are key highlights to assist in your post-rebuttal discussion:

&nbsp;

> **1. Dealing with non-stationary TS (EHZC, mvHV)**

EHZC and mvHV expressed doubt about the ability to capture non-stationarity of TS. We emphasize that **the primary focus on this work is on improving CL for TS** (which is SOTA in TS representation learning), **rather than addressing specific problems in TS like non-stationarity**. Indeed, our proposed **SoftCLT is a simple** yet effective soft contrastive learning strategy for TS, so addressing such problems at the expense of simplicity might result in hampering our main contribution.


Rather, such specific problems can be addressed by other strategies, and our method can be applied on top of them to improve the performance further. For example, we applied our SoftCLT to CoST (a previous method designed to learn disentangled seasonal-trend representations) and observed performance gain, which can be found in the new experiment in **Table H.3 in Appendix H**.


Nonetheless, we conducted additional experiments in **Appendix M** to demonstrate that our soft CL can address these issues. **Figures M.1 and M.2** demonstrate that soft CL effectively handles TS with seasonality and distribution shift, respectively, which hard CL cannot address.

&nbsp;

> **2. Applying SoftCLT to CoST (bKG2)**

As CoST exhibits SOTA performance in time series forecasting tasks, we applied our method to CoST and showed the results in **Table H.3**. It shows consistent improvements across various datasets and horizons.

&nbsp;


> **3. CL methods in medical domain (e9k3)**

e9k3 expressed doubt about the absence of related works regarding CL in the medical domain. We have addressed this concern by incorporating these works in **Section 2**.

&nbsp;

> **4. Different kernels for soft assignments (e9k3)**

e9k3 suggested using different kernels for assigning soft assignments in CL. In addition to the Gaussian function, we conducted additional experiments using the Laplacian kernel for instance-wise CL, as suggested by the reviewer. The results, presented in **Table F.1**, demonstrate that using the sigmoid function yields better performance.

&nbsp;

> **5. Minor Issues**

Clarity (e9k3)
- e9k3 concerned about clarity in some parts, and in response, we have added the definitions of instance-wise and temporal CL in **Section 1** and **Figure 1**.

Explanation of augmentations (EHZC)
- In response to concerns raised by EHZC about data augmentation, we've added details in **Section 4**. Note that our method is not constrained to specific data augmentation, and we followed the augmentation techniques used in the original methods.

Complexity issue of DTW (mvHV)
- mvHV expressed concerns about the complexity of DTW. We emphasize that DTW is just one choice for our distance metric, and other metrics are possible. As shown in **Table 6(d)**, DTW shows the best performance but other simpler metrics also show decent performance. Moreover, the pairwise distance matrix of DTW can be precomputed offline or cached for efficiency, and FastDTW can be used as a fast approximation with negligible errors.

Soft assignment's dependency on the scale of the input data (mvHV)
- mvHV raised concerns that the scale of the input data affects the resulting soft assignments. We initially omitted details regarding the min-max normalization process of the pairwise distance matrix; this normalization makes the soft assignment independent to the scape of the input data. In **Section 3.2**, we have clarified that the distance is min-max normalized.

---

### Public Comment · ~Huy_Mai3 · 2025-12-12
**Clarification on Notation in Section 3.2**

Dear Authors,

Thank you for this very interesting work on Soft Contrastive Learning. The method for incorporating soft assignments based on data-space similarity is insightful.

I have a small question regarding the notation in Section 3.2 that I hope you can clarify to aid my understanding. In the paragraph just before Equation (2), the text states:

> "Let $r_{i,t}$ = $r_{i+2N,t}$ and $\tilde r_{i,t} = r_{i+N,t}$ be the embedding vectors from two augmentations of $x_i$ at timestamp $t$ for conciseness."

I had two points of confusion regarding this sentence:
1. Regarding $\tilde r_{i,t} = r_{i+N,t}$: While this defines the positive pair, the $\tilde r$ notation itself does not appear to be used in the subsequent loss equations, which instead use explicit indexing like (i, i+N). I just wanted to confirm if this notation was for illustrative purposes or if I might have missed its direct usage elsewhere.
2. Regarding $r_{i,t}$ = $r_{i+2N,t}$: I was slightly puzzled by this definition. In a setup with a batch of N samples augmented into 2N views, an index of i+2N seems to be out of bounds. I was wondering if this might be a typo and was intended to refer to the anchor view, $r_{i,t}$? Or perhaps it relates to a specific implementation detail?

Any clarification you could provide would be immensely helpful for my understanding of the instance-wise loss formulation.

Thank you for your time and for the great contribution.

Best regards,

A fellow researcher

---

### Meta-Review · Program_Chairs · 2023-12-05

**Metareview:**

The manuscript presents soft CLT - a soft constrastive learning algorithm for time-series - the soft assignments include instant-wise and temporal contrastive losses. It has been demonstrated that the softCLT is applicable to a number of frameworks such as in semi-supervised classification, transfer learning and anomaly detection, in addition to supervised classification (I think there is also a potential to apply to continual learning paradigms). It is also applicable to a wide variety of augmentation techniques. Furthermore, authors have also shown that the soft CL has the potential to effectively handle distribution shifts prevalent in TS (which hard CL can not).

Performance and ablation studies are sufficient (Authors have compared with CoST, as suggested by Reviewer bKG2. Furthermore, sufficient justifications are added based on the reviewers comments. The manuscript is well written and provides a novel contrastive learning algorithm for TS.

Although the effect of noise in data is not considered, it is understood that the component is beyond the scope of the current work.

**Justification For Why Not Higher Score:**

The reviewers point out missing comparisons with related works, as well as some clarity issues.

**Justification For Why Not Lower Score:**

The manuscript is well-written and warrants a good discussion at ICLR 2024.

It has been demonstrated that the softCLT is applicable to a number of frameworks such as in semi-supervised classification, transfer learning and anomaly detection, in addition to supervised classification (I think there is also a potential to apply to continual learning paradigms).

It is also applicable to a wide variety of augmentation techniques.

Furthermore, authors have also shown that the soft CL has the potential to effectively handle distribution shifts prevalent in TS (although it is not clear what kind of distributional/seasonal shifts can be handled).

---

### Decision · Program_Chairs · 2024-01-16

Accept (spotlight)